# Detection and variability of combustion-derived vapor in an urban basin

Richard P. Fiorella[1], Ryan Bares[2], John C. Lin[2,3], James R. Ehleringer[4,3], and Gabriel J. Bowen[1,3]

[1]Department of Geology and Geophysics, University of Utah, 115 S 1460 E, Salt Lake City, UT, 84112, USA
[2]Department of Atmospheric Sciences, University of Utah
[3]Global Change and Sustainability Center, University of Utah
[4]Department of Biology, University of Utah

*Correspondence to:* Richard Fiorella (rich.fiorella@utah.edu)

**Abstract.** Water emitted during combustion may comprise a significant portion of ambient humidity (>10%) in urban areas, where combustion emissions are strongly focused in space and time. Stable water vapor isotopes can be used to apportion measured humidity values between atmospherically transported and combustion-derived water vapor, as combustion-derived vapor possesses an unusually negative deuterium excess value (d-excess, $d = \delta^2 H - 8\delta^{18}O$). We investigated the relationship between the d-excess of atmospheric vapor, ambient $CO_2$ concentrations, and atmospheric stability across four winters in Salt Lake City, UT. We found a robust inverse relationship between $CO_2$ excess above background and d-excess on sub-diurnal to seasonal timescales, which was most prominent during periods of strong atmospheric stability that occur during Salt Lake City winter. Using a Keeling-style mixing model approach, and assuming a molar ratio of $H_2O$ to $CO_2$ in emissions of 1.5, we estimated the d-excess of combustion-derived vapor in Salt Lake City to be $-179 \pm 17‰$, consistent with the upper limit of theoretical estimates. Based on this estimate, we calculate that vapor from fossil fuel combustion often represents 5-10% of total urban humidity, with a maximum estimate of 16.7%, consistent with prior estimates for Salt Lake City. Moreover, our analysis highlights that changes in the observed d-excess during periods of high atmospheric stability cannot be explained without a vapor source possessing a strongly negative d-excess value. Further refinements in this humidity apportionment method, most notably empirical validation of the d-excess of combustion vapor or improvements in the estimation of the background d-excess value in the absence of combustion, can yield more certain estimates of the impacts of fossil fuel combustion on urban humidity and meteorology.

## 1 Introduction

Fossil fuel combustion releases carbon dioxide and water to the atmosphere. Annual carbon emissions are estimated to be 9.4 $Pg\,C\,y^{-1}$ (Le Quéré et al., 2018), which suggests annual water emissions from combustion of 21.1 Pg, assuming a mean molar emissions ratio between $H_2O:CO_2$ of 1.5 (section 2, and also Gorski et al., 2015). This water flux is negligible in the hydrologic cycle on global and annual timescales (e.g., Trenberth et al., 2006), but it may be significant to urban hydrologic cycling and meteorology as fossil fuel emissions are tightly concentrated in space and time (Bergeron and Strachan, 2012; Duren and Miller, 2012; Gorski et al., 2015; Sailor, 2011; Salmon et al., 2017). In turn, water vapor from fossil fuel combustion may

impact urban air quality and meteorology, for example, through direct changes in radiative balance by increased water vapor concentrations (Holmer and Eliasson, 1999; McCarthy et al., 2010), impacts on aerosols and cloud properties (Pruppacher and Klett, 2010; Mölders and Olson, 2004; Kourtidis et al., 2015; Twohy et al., 2009; Carlton and Turpin, 2013; Kaufman and Koren, 2006), and altered local or downwind precipitation amounts (Rosenfeld et al., 2008). Where combined with atmospheric stratification, these changes can potentially lengthen or intensify periods of elevated particulate pollution in cities, which would directly impact public health through increased incidence of acute cardiovascular (Morris et al., 1995; Brook et al., 2010) or respiratory (Dockery and Pope, 1994) illness. However, using standard meteorological measurements it remains difficult to isolate combustion-derived vapor (CDV) from "naturally-occurring" water vapor, or vapor from other anthropogenically-influenced fluxes (e.g., snow sublimation from buildings), making the impact of CDV on the urban atmosphere difficult to assess.

Stable water vapor isotopes represent a promising method to partition observed water vapor between combustion and advection sources (Gorski et al., 2015). Combustion of hydrocarbons produces water from the reaction of atmospheric oxygen, which is $^{18}O$-enriched relative to the international standard, Vienna Standard Mean Ocean Water (VSMOW) (+23.9 ‰, Barkan and Luz, 2005), and structurally-bound fuel hydrogen, which is $^{2}H$-depleted relative to VSMOW due to preference for $^{1}H$ over $^{2}H$ during biosynthetic reactions (e.g., Estep and Hoering, 1980; Sessions et al., 1999). The reaction of $^{18}O$-enriched oxygen with $^{2}H$-depleted fuels produces vapor with an unusually negative deuterium excess value ($d = \delta^2 H - 8\delta^{18}O$; Dansgaard, 1964) that is distinct compared to d-excess value in the "natural" hydrological cycle. Deuterium excess is $\sim 10$ ‰, on average, in precipitation (Dansgaard, 1964; Rozanski et al., 1993), and ranges in "natural" waters from +150-200 ‰ in vapor in the upper troposphere (Blossey et al., 2010; Bony et al., 2008; Webster and Heymsfield, 2003) to $\sim -60$‰ in highly evaporated surface waters (e.g., Fiorella et al., 2015). In contrast, Gorski et al. (2015) estimated CDV d-excess values for fuels in Salt Lake Valley (SLV) ranging from $-180$ to $-470$ ‰, depending on the isotopic composition of the fuel and the degree of equilibration of oxygen isotopes between $CO_2$ and $H_2O$ in combustion emissions.

The Salt Lake City, UT metro area (population of $\sim 1.15$ million) is located within the SLV. The SLV ($\sim 1300 - 1500$ m) is bounded on the west by the Oquirrh Mountains ($\sim 2200 - 2500$ m), on the east by the Wasatch Mountains (>3000 m), and on the south by the Traverse Mountains ($< 2000$ m). The northwest corner of the basin is bounded by the Great Salt Lake. During the winter, cold air often pools in the SLV, increasing atmospheric stability and limiting transport of combustion products away from the city and impairing air quality. Previous work in the SLV indicated that CDV comprised up to $\sim 13\%$ of urban specific humidity during strong inversion events in winter 2013-2014 (Gorski et al., 2015). Here we combine those data with three additional winters of water vapor isotope measurements in Salt Lake City, UT (DJF 2014-2017) to refine our estimate of the d-excess of CDV, update estimates of the contributions of CDV to the urban atmosphere, and identify the largest sources of error that can be addressed or reduced in future studies.

## 2 Stoichiometric relationships between $CO_2$ and CDV and fuel use in SLV

The ratio of $CO_2$ to CDV in fossil fuel emissions depends on the stoichiometry of the fuels used. The chemical reaction for the idealized combustion of a generic hydrocarbon is:

$$C_xH_y + (x + y/4)O_2 \rightarrow xCO_2 + (y/2)H_2O \tag{R1}$$

5  The molar ratio of $H_2O$ and $CO_2$ in product vapor is defined here as the emissions factor ($ef$), and arises directly from the molar ratio of hydrogen and carbon in the fuel as $y/2x$. Of simple hydrocarbons, methane ($CH_4$) has the greatest $ef$ value of 2. Longer-chained hydrocarbons, such those in gasoline, have lower $ef$ values. Octane ($C_8H_{18}$) has an $ef$ value of 1.125, for example (Gorski et al., 2015).

Fuels burned within the SLV are generally petroleum products and natural gas, with the latter being extensively used in the 10  winter for residential heating. Seasonal patterns of fuel use emerge from both "top-down" and "bottom-up" style emissions estimates. A high-resolution, bottom-up, building-level emissions inventory has been produced for Salt Lake County as part of the HESTIA project (Gurney et al., 2012; Patarasuk et al., 2016; Zhou and Gurney, 2010). On an annual basis, onroad transport represents 42.9% of Salt Lake County emissions, followed by the residential (20.8%) and industrial (12.6%) sectors (Patarasuk et al., 2016). The commercial, electric generation, and non-road transport sectors comprise the remaining 23.7% 15  of Salt Lake County emissions. In winter, however, the residential sector is a much larger contributor to Salt Lake County emissions (34.4%), followed by the onroad transport (34.3%) and commercial sectors (13.1%) (Table 1). The remaining 18.2% of emissions arise from the non-road transport, electricity production, and industrial sectors. The increased prominence of residential and commercial sector emissions during the winter, primarily at the expense of onroad and industrial emissions, likely results from a greater heating demand and a concomitant increase in natural gas use. "Top-down" observations of stable 20  carbon isotope compositions in atmospheric $CO_2$ in the SLV reflect this seasonal change in carbon inputs from primarily from gasoline combustion and respiration in the summer to a much stronger signal from natural gas in the winter (Pataki et al., 2003, 2005).

From these considerations, we estimate a valley-scale $ef$ value using the HESTIA emissions inventory (Patarasuk et al., 2016) and appropriate emissions factors for natural gas, petroleum, and sub-bituminous coal resources. Natural gas was as-25  sumed to be composed of 90% methane, 8% ethane, and 2% propane (Schobert, 2013), yielding an $ef$ value of 1.95. Petroleum products such as gasoline, jet fuel, and fuel oil, were assumed to be 85% C and 15% H by mass (Schobert, 2013; Dabelstein et al., 2012), yielding an $ef$ value of 1.05. Finally, an $ef$ value of 0.5 was assigned to coal, assuming a molar ratio of hydrogen to carbon of 1 (Schobert, 2013). Fuels or fuel mixtures were assigned to each economic sector in the HESTIA data set (Table 1). Mobile emissions (airport, on road, non-road, and railroad) were assigned petroleum sources, while the residential and elec-30  tricity generation sectors were assigned natural gas sources (Table 1). Coal combustion supplies the majority of electricity in Utah and in SLV, but the power plants supplying the SLV are outside of the valley to the south. Electricity generation facilities within the SLV are primarily natural gas facilities. Commercial and industrial source emissions were apportioned using the state-wide ratios of carbon emissions across fuel sources for these economic sectors collected by the US Energy Information Administration (EIA, 2015). Commercial sector emissions were assumed to be 83.3% natural gas and 16.7% petroleum, while

**Table 1.** HESTIA Emissions Estimates and estimated $ef$ values for Salt Lake County

| Economic sector | December | January | February | DJF Sum | Natural Gas | Petroleum | Coal | estimated $ef$ |
|---|---|---|---|---|---|---|---|---|
| | (Gg C) | | | | (%) | | | |
| Airport | 8.47 | 8.74 | 8.04 | 25.24 | 0.0 | 100.0 | 0.0 | 1.05 |
| Commercial | 45.30 | 47.47 | 35.16 | 127.92 | 83.3 | 16.7 | 0.0 | 1.80 |
| Electricity Generation | 10.01 | 6.50 | 6.84 | 23.36 | 100.0 | 0.0 | 0.0 | 1.95 |
| Industry | 33.21 | 33.81 | 33.21 | 100.24 | 46.7 | 35.1 | 18.2 | 1.37 |
| Non-road | 8.90 | 8.59 | 8.93 | 26.42 | 0.0 | 100.0 | 0.0 | 1.05 |
| Onroad | 113.50 | 113.41 | 108.94 | 335.85 | 0.0 | 100.0 | 0.0 | 1.05 |
| Railroad | 1.17 | 1.17 | 1.06 | 3.40 | 0.0 | 100.0 | 0.0 | 1.05 |
| Residential | 116.14 | 125.64 | 94.48 | 336.26 | 100.0 | 0.0 | 0.0 | 1.95 |
| Weighted average $ef$ | 1.52 | 1.53 | 1.48 | 1.51 | | | | |

industrial emissions were assumed to arise from a combustion mixture of 46.8% natural gas, 35.1% petroleum, and 18.1% coal (Table 1). Weighting these economic sectors and fuel sources by their relative emissions amounts yields a Salt Lake County scale estimate of $ef$ of 1.51 for winter, with individual months ranging from 1.48 to 1.53. Based on this analysis, we consider an estimate for $ef$ of 1.5 going forward.

## 3  Methods

### 3.1  Estimates of Atmospheric Stratification

The SLV experiences periods of enhanced atmospheric stability each winter when cold air pools in the valley under warmer air aloft (Lareau et al., 2013; Whiteman et al., 2014). Atmospheric stratification is present when potential temperature increases with height. Nocturnal stratification is common in many settings due to more rapid radiative cooling near the surface than aloft, but the SLV and other topographic basins can experience periods of extended atmospheric stability lasting longer than a diurnal cycle (Lareau et al., 2013; Whiteman et al., 2001, 1999). These periods are commonly referred to as persistent cold air pools (PCAPs) (Gillies et al., 2010; Green et al., 2015; Malek et al., 2006).

We assess large-scale SLV vertical stability using twice-daily atmospheric soundings from the Salt Lake City Airport (KSLC, 0 and 12 UTC, or 5 and 17 LT). Sounding profiles were obtained from the Integrated Global Radiosonde Archive (IGRA) (Durre and Yin, 2008), and interpolated to $10\,\mathrm{m}$ resolution between the surface ($\sim 1290\,\mathrm{m}$) and $5{,}000\,\mathrm{m}$. We calculate two metrics of atmospheric stability from the radiosonde data: a bulk valley heat deficit (VHD) and an estimated mixing height. The VHD is the energy that must be added between the surface and some height to bring this portion of the atmosphere to the dry adiabatic lapse rate (e.g., $\frac{\partial \theta}{\partial z} = 0.0\,\mathrm{K\,km^{-1}}$ or $\frac{\partial T}{\partial z} = -9.8\,\mathrm{K\,km^{-1}}$). VHD is calculated following prior studies of winter stability in the

SLV (Baasandorj et al., 2017; Whiteman et al., 2014):

$$VHD = c_p \sum_{1290 \text{ m}}^{2200 \text{ m}} \rho(z)[\theta_{2200 \text{ m}} - \theta(z)]\Delta z \tag{1}$$

where $c_p$ is the specific heat capacity at constant pressure for dry air (1005 J kg$^{-1}$ K$^{-1}$), $\rho(z)$ is the air density as a function of height (kg m$^{-3}$), $\theta_{2200 \text{ m}}$ and $\theta(z)$ are the potential temperatures at 2200 m above sea level and at height $z$ respectively (K), and $\Delta z$ is the thickness of each layer (10 m). The upper bound in the VHD calculation (2200 m) is determined by the elevation of the Oquirrh Mountain ridgeline, which forms the western valley boundary. Following Whiteman et al. (2014), we define a PCAP as three or more consecutive soundings with a VHD >4.04 MJ m$^{-2}$. This VHD threshold of 4.04 MJ m$^{-2}$ corresponds to the mean VHD in days where the SLV daily fine particulate matter concentration (PM$_{2.5}$) exceeds half of the US National Ambient Air Quality Standard for PM$_{2.5}$ (17.5 μg m$^{-3}$) (Whiteman et al., 2014), and has been used in subsequent studies of SLV air quality and atmospheric stability (Baasandorj et al., 2017; Bares et al., 2018). We have retained this convention for intercomparison with prior studies.

Mixing height estimates depend on whether a surface-based temperature inversion is present or absent. If the sounding features an surface-based inversion, the mixing height is estimated as the height at the top of the surface-based inversion (Bradley et al., 1993). If there is no surface-based inversion, the mixing height is estimated using a bulk Richardson number method (Vogelezang and Holtslag, 1996; Seidel et al., 2012). The bulk Richardson number, which is a measure of the ratio of buoyancy to shear production of turbulence, is calculated as:

$$Ri(z) = \frac{(g/\theta_{vs})(\theta_v(z) - \theta_{vs})(z - z_s)}{(u(z) - u_s)^2 + (v(z) - v_s)^2 + bu_*^2} \tag{2}$$

where $Ri(z)$ is the bulk Richardson number as a function of height, $g$ is the acceleration due to gravity (9.81 m s$^{-2}$), $\theta_v$ is the virtual potential temperature (K), $z$ is the altitude (m above sea level), $u$ and $v$ are the zonal and meridional wind components (m s$^{-1}$), and $bu_*^2$ is the effect of surface friction. A subscript 's' indicates these are surface values. As $u_*$ is not available from radiosonde observations, we assumed frictional effects were negligible (Seidel et al., 2012). This assumption is particularly well justified during stable atmospheric conditions (Vogelezang and Holtslag, 1996), such as during PCAPs. The mixing height was identified as the lowest altitude where $Ri(z)$ was greater than a critical value of 0.25.

## 3.2 Water Vapor Isotope Data

Water vapor isotope data were collected using a Picarro L2130-i water vapor isotope analyzer (Santa Clara, CA). Vapor was sampled from the roof of the eight-story ($\sim 35$ m above the ground) William Browning Building on the University of Utah campus (UOU, 40.7662°N, 111.8458°W, 1440 m above sea level) through copper (prior to winter 2016/2017) or teflon tubing, using a diaphragm pump operating at $\sim 3$ L min$^{-1}$. Standards were analyzed every 12 hours using the Picarro Standards Delivery Module (Table 2), using lab air pumped through a column of anhydrous calcium sulfate (Drierite) as a dry gas source. We calibrated the data using the University of Utah vapor processing scripts, version 1.2. Calibration of raw instrument values at $\sim 1$ Hz on the instrument scale to hourly averages on the VSMOW scale proceeds across three stages: (1) Measured isotope

**Table 2.** Laboratory Standard Isotopic Compositions

|  | Light Standard | | Heavy Standard | |
|  | $\delta^{18}O$ | $\delta^2H$ | $\delta^{18}O$ | $\delta^2H$ |
| --- | --- | --- | --- | --- |
| Prior to 16 Feb 2017 | −16.0 | −121.0 | −1.23 | −5.51 |
| After 16 Feb 2017 | −15.88 | −119.66 | 1.65 | 16.9 |

values are corrected for an apparent dependence on cavity humidity, using correction equations developed by operating the standards delivery module at a range of injection rates, corresponding to cavity humidity values of 500-30000 ppm. Instrumental precision is determined in this step, with uncertainties arising both from a decrease in instrument precision with decreasing cavity humidity, and uncertainty in the regression equation to correct for this bias. The humidity correction is determined by a linear regression of the deviation of isotopic composition from the measured isotopic composition at a reference humidity against the inverse of cavity humidity. The reference humidity used is 15,000-25,000 ppm, a range where the instrument response is linear and at which liquid water samples are measured and lab standards are calibrated. Additional details on this correction are provided in a supplement. (2) Analyzer measurements are calibrated to the VSMOW-VSLAP scale using two standards of known isotopic composition delivered by the standards delivery module (Table 2), using calibration periods that bracket a series of ambient vapor measurements to correct for analytical drift, (3) corrected measurements were aggregated to an hourly time step. Measurement uncertainties are primarily limited by changes in instrument precision with cavity humidity, and $1\sigma$ uncertainties range from 0.88‰ for $\delta^{18}O$, 3.61‰ for $\delta^2H$, and 7.93‰ for d-excess (assuming error independence) at a humidity of 1000 ppm; to 0.14‰ for $\delta^{18}O$, 0.53‰ for $\delta^2H$, and 1.24‰ for d-excess at a humidity of 10000 ppm.

### 3.3 $CO_2$ and meteorological measurements

Meteorological measurements were co-located with water vapor isotope sampling on the roof of the UOU. Temperature, humidity, wind speed, solar radiation, and pressure measurements are all made at 5-min averages (Horel et al., 2002), and were averaged to 1 hour blocks for analysis.

$CO_2$ measurements were made in two different locations during the study period. Prior to August 2014, $CO_2$ measurements were made on the roof of the Aline Skaggs Biology Building (ASB) on the University of Utah campus, $\sim 0.25$ km south of the William Browning Building (coded as UOU). $CO_2$ and $H_2O$ measurements made at ASB were performed using a Li-Cor 7000. Atmospheric air was drawn through a 5 L mixing volume and measured every five minutes. Pressure and $H_2O$ dilution corrections were applied by the Li-Cor. All measurements were recorded to a Campbell Scientific CR23X.

From August 2014 onwards, $CO_2$ measurements have been made at the UOU where they are co-located with meteorological measurements and the water vapor isotope described in section 3.2. Atmospheric $CO_2$, $CH_4$ and $H_2O$ measurements were performed using a Los Gatos Research Off-Axis Integrated Cavity Output Spectroscope (Model 907-0011, Los Gatos Research Inc., San Jose, CA). Measurements were recorded at 0.1 Hz. The effects of water vapor dilution and spectrum broadening (Andrews et al., 2014) were corrected by LGR's real-time software, and were independently verified through laboratory testing.

At both ASB and UOU, calibration gases were introduced to the analyzer every three hours using three whole-air, dry, high-pressure reference gas cylinders with known $CO_2$ concentrations, tertiary to the World Meteorological Organization X2007 $CO_2$ mole fraction scale (Zhao and Tans, 2006). Concentrations of the calibration gases spanned the expected range of atmospheric observations. Each standard of known concentration is linearly interpolated between two consecutive calibration periods to represent the drift in the averaged measured standards over time. Ordinary least squares regression is then applied to the interpolated reference values during the atmospheric sampling periods to generate slope and intercept estimates. These are then used to correct all uncalibrated atmospheric observations between calibration periods. Analytical precision is estimated to be $\sim 0.1$ ppm.

Seven months of overlapping data were collected at both ASB and UOU and analyzed to identify any significant difference in measurement locations. The two locations are highly similar ($CO_{2,UOU} = 0.98CO_{2,ASB} + 8.087, r^2 = 0.96$), though pollutants appear to "mix-out" at the end of a PCAP event approximately one hour earlier at ASB relative to UOU. We do not adjust the ASB time series as the potential time shift is small, and the period of overlapping records is short and does not span a full annual cycle.

## 3.4 Mixing analysis between meteorological humidity and combustion-derived vapor

CDV can be assessed by considering a two-part isotopic mixing model that treats meteorological or advected vapor and CDV as the end members. We develop a schematic demonstrating the 'natural' evolution of d-excess under atmospheric moistening and condensation conditions, as well as through moistening via the addition of CDV. The isotopic composition of an air parcel losing moisture in a Rayleigh condensation process can be modeled as (Gat, 1996):

$$\delta = \left[ \left( \delta_0 + 1 \right) \left( \frac{q}{q_0} \right)^{\alpha - 1} - 1 \right] \tag{3}$$

where $\delta$ is the isotopic composition, $q$ is the specific humidity, and $\alpha$ is the temperature-dependent equilibrium fractionation factor between vapor and the condensate. A subscript zero indicates the initial conditions of a parcel prior to condensation. Humidity is removed from the air parcel through adiabatic cooling starting from the parcel's initial dew point temperature and cooling in 0.5 K intervals to 243 K; progressive cooling is used to account for changes in $\alpha$ with temperature. $\delta^{18}O$ and $\delta^{2}H$ are modeled separately and then combined to estimate the evolution of d-excess throughout condensation. We used fractionation factors for vapor over liquid for temperatures above 273 K (Horita and Wesolowski, 1994) and for vapor over ice for temperatures below 253 K (Majoube, 1970; Merlivat and Nief, 1967). We interpolated $\alpha$ values between 273 K and 253 K to account for mixed-phase processes between these temperatures. As the heavy isotopes of both oxygen and hydrogen are progressively removed through condensation, d-excess increases as humidity is decreased, approaching a limit of 7000‰ if all $^{2}H$ and $^{18}O$ were removed (Bony et al., 2008).

We also modeled the isotopic evolution of d-excess in an air parcel in the absence of CDV experiencing mixing between the moist and dry end members of the Rayleigh distillation curve. D-excess is modeled throughout this humidity range as a

mass-weighted mixing model average of the d-excess values of both end members:

$$d_{\text{mix}} = \frac{d_{\text{dry}}q_{\text{dry}} + d_{\text{moist}}q_{\text{moist}}}{q_{\text{dry}} + q_{\text{moist}}} \qquad (4)$$

Likewise, moistening of the lower troposphere by CDV can be modeled as a mixing process between CDV and the background "natural" water vapor:

$$d_{\text{mix}} = \frac{d_{\text{CDV}}q_{\text{CDV}} + d_{\text{bg}}q_{\text{bg}}}{q_{\text{mix}}} \qquad (5)$$

where subscripts CDV, bg, and mix refer to properties of CDV, the atmospheric moisture in the absence of CDV, and values of the mixed parcel, respectively. Gorski et al. (2015) assumed a mean value of $-225‰$ for $d_{\text{CDV}}$ based on a few direct measurements. Adopting this value, we construct a model framework to explain changes in d-excess relative to humidity expected from natural condensation and mixing pathways as well as the addition of moisture via CDV (Fig. 1), but also revisit

this assumption based on further analysis of our data (below). Drying the atmosphere by mixing in a dry air mass in the absence of CDV or by Rayleigh condensation increases the d-excess of ambient vapor, whereas atmospheric moistening occurring due to mixing with a moist air mass can decrease the d-excess of ambient vapor. The response of d-excess due to these natural processes is non-linear with respect to changes in humidity, and very similar between condensation and mixing of "natural" air masses (Fig. 1). In contrast, small mass additions of CDV (up to 500 ppm) produce a strong, quasi-linear decrease in $d_{\text{mix}}$

with increasing $q_{\text{CDV}}$ (Fig. 1). Assuming a representative $ef$ value of 1.5 (section 2), 100 or 500 ppm of CDV correspond to $CO_2$ increases of 66.7 or 333.3 ppm, respectively. Deviation from the "natural" air mass mixing line is greatest at low $q_{\text{bg}}$ for a given $q_{\text{CDV}}$, as CDV comprises a larger fraction of $q_{\text{mix}}$.

Recasting these mixing-model equations following the Miller-Tans (2003) formulation of the Keeling (1958; 1961) mixing model, we can estimate $d_{\text{CDV}}$. In this framework, the product of observed d and q (e.g., $d_{\text{obs}}$ and $q_{\text{obs}}$) is proportional to $q_{\text{CDV}}$:

$$d_{\text{obs}}q_{\text{obs}} = d_{\text{CDV}}q_{\text{CDV}} + d_{\text{bg}}q_{\text{bg}} \qquad (6)$$

If we assume that $q_{\text{CDV}}$ is linearly related to the increase in $CO_2$ above background concentrations, $d_{\text{CDV}}$ can be estimated as the slope of a linear regression between $d_{obs}q_{obs}$ and observed $CO_2$ concentrations:

$$d_{\text{obs}}q_{\text{obs}} = d_{\text{CDV}}(ef)[CO_2 - \min(CO_2)] + d_{\text{bg}}q_{\text{bg}} \qquad (7)$$

where $ef$ is the emissions factor, which is the stoichiometric ratio of $H_2O$ to $CO_2$ in combustion products, and $[CO_2 -$

$\min(CO_2)]$ represents the amount of excess $CO_2$ in the atmosphere above the background value. The $ef$ parameter depends on the molar ratios of hydrogen to carbon in the fuel source; we estimate an fuel-source-weighted SLV-scale $ef$ value for winter of 1.5, but note that $ef$ values for hydrocarbon fuels can vary from $< 0.5 - 2$. We define the background $CO_2$ value, $\min(CO_2)$, to be the seasonal minimum value observed at the UOU or the ASB. Observations of urban $\delta^{13}C$-$CO_2$ and atmospheric modeling of the SLV indicate that wintertime increases in $CO_2$ above background concentrations are driven by anthropogenic emissions,

and that the contribution from local respiration to urban $CO_2$ enhancement is likely negligible (Pataki et al., 2003, 2005, 2007; Strong et al., 2011). We apply two linear mixed models where PCAP-to-PCAP event-scale variability is treated as a random

effect to estimate $d_{\mathrm{CDV}}$: in the first, the slope is assumed to be constant across all PCAP events but the intercept is allowed to vary, while in the second, both the slope and intercept are allowed to vary across PCAP events. These models are constructed to find the best-fit slope, and therefore the best-fit estimate of $d_{\mathrm{CDV}}$, across all PCAP events. As a result, they implicitly assume that changes in $d_{\mathrm{CDV}}$ through time are small compared to changes in $d_{\mathrm{bg}}q_{\mathrm{bg}}$, or that changes in the emissions profile of the SLV are small compared to environmental variability in humidity and d-excess. We consider only the second model in our results as we find it has more support than the first model, with this selection determined based on lower Akaike and Bayesian Information Criteria (AIC and BIC) for the second model. AIC and BIC are both model selection tools that optimize model parsimony by evaluating a model's likelihood against a penalty based on the number of model parameters.

Finally, the fraction of urban humidity comprised of CDV can be estimated by solving equation 6 for $q_{CDV}/q_{obs}$ using the constraint that $q_{obs} = q_{CDV} + q_{bg}$:

$$\frac{q_{CDV}}{q_{obs}} = \frac{d_{obs} - d_{bg}}{d_{CDV} - d_{bg}} \tag{8}$$

Using this equation, we estimate a maximum contribution of CDV to boundary layer humidity for each PCAP where water isotope data are available using the minimum $d_{obs}$ value from each PCAP. We assume a constant value of $d_{CDV}$, determined from the slope of the linear mixed model described above. Two estimates of $d_{bg}$ were made for each PCAP based on the assumptions that $d_{bg}$ reflects: (a) the mean observed $d$ value for the 12 hours prior to the initiation of the PCAP, or (b) the mean $d$ value for the 12 hour period where the 12 hour moving average $CO_2$ concentration falls below 415 ppm. For (b), if the 12 hour average $CO_2$ concentration fails to fall below 415 ppm between two PCAPs, $d_{bg}$ is estimated from the minimum $CO_2$ value between these PCAP events.

## 4 Results

We observed 26 PCAP events across four winters, with seven, four, seven, and eight occurring during DJF 13/14, 14/15, 15/16, and 16/17, respectively (Fig. 2). VHD exceeded $4.04\,\mathrm{MJ\,m^{-2}}$ for a 30%, 18%, 27%, and 25% of the observed KSLC soundings during each winter. Variability of 1 to $2\,\mathrm{MJ\,m^{-2}}$ between consecutive soundings is common, and results from the diurnal cycle of surface heating during the day and radiative cooling at night (Whiteman et al., 2014). Calculated mixing heights ranged from the surface (0 m AGL) to 3390 m AGL, with a median value of 270 m AGL. The mean mixing height and its variance are low in December and January, though both increase in February as solar radiation increases and more energy is available to grow the daytime convective boundary layer.

$CO_2$ concentrations show close inverse associations with measured d-excess values across diurnal to synoptic timescales (Fig. 3). Paired d-excess and $CO_2$ measurements are available for 76.8% of the period of record, including for 22 of the 26 PCAP events. $CO_2$ concentrations and d-excess values were inversely cross-correlated for all four winter periods ($r = -0.589$, $-0.547$, $-0.428$, and $-0.527$ for each consecutive winter). The maximum cross-correlation was observed with zero lag in DJF 14/15 and 16/17, whereas d-excess lagged $CO_2$ by 1 hour in DJF 13/14 and 15/16. For each winter season, minimum/maximum hourly $CO_2$ concentrations were 397/637 ppm, 400/581 ppm, 404/598 ppm, 406/653 ppm, whereas minimum/maximum hourly d-excess values were $-26.4/24.5\,\text{‰}$, $-10.5/19.4\,\text{‰}$, $-8.0/12.9\,\text{‰}$, and $-26.8/14.3\,\text{‰}$.

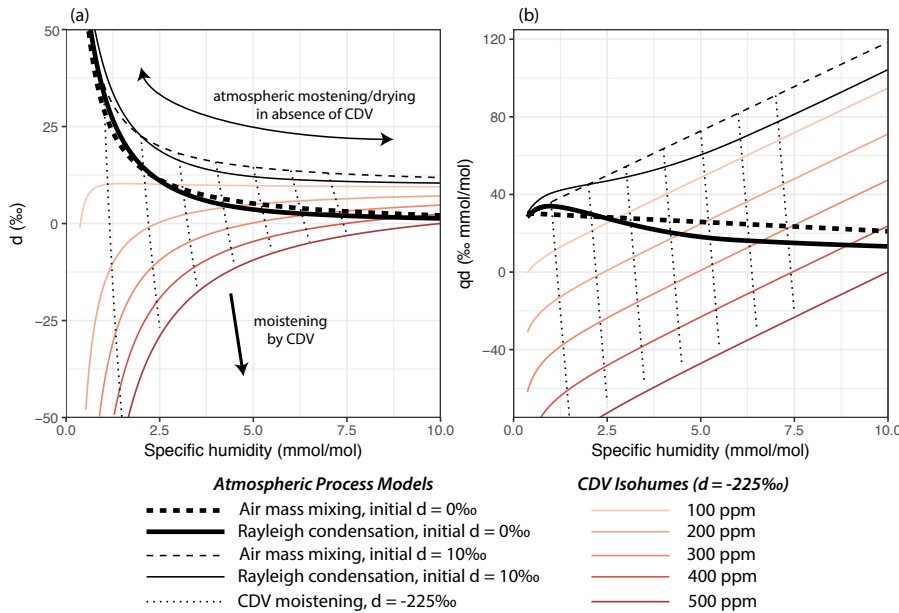

**Figure 1.** Schematic of expected changes in the d-excess of atmospheric vapor with changes in humidity associated with atmospheric moistening and drying in the absence of CDV due to Rayleigh distillation (solid black lines) or air mass mixing (dashed black lines) or the addition of CDV (dotted black lines). Models for Rayleigh distillation and air mass mixing are shown for two initial d-excess values of the moist end member: 0‰ (thick lines) and 10‰ (thin lines). Panel (a) shows this relationship of $d$ (‰) vs specific humidity, $q$ (mmol mol$^{-1}$), where mixing processes trace hyperbolic pathways, and panel (b) shows the same models but with axes of $qd$ (‰ mmol mol$^{-1}$) against $q$ (mmol mol$^{-1}$), where mixing processes are linear. Finally, lines across a red gradient are drawn to show the impact of fixed amounts of CDV addition ranging from 100 ppm (light) to 500 ppm (dark) as a function of specific humidity.

During each PCAP event, $CO_2$ was elevated relative to its background value. For most PCAP events, d-excess decreased commensurately with the increase in $CO_2$; however, several exceptions were observed. For example, PCAPs in February 2016 and 2017 showed diurnal cyclicity in d-excess and $CO_2$ during the event, but these periods often exhibited a multiday period of $CO_2$ increase and d-excess decrease prior to atmospheric stability reaching the VHD threshold for a PCAP. In these events, the

5    bulk of the d-excess decrease occurs prior to the onset of the PCAP as defined by the VHD metric, and d-excess exhibits strong diurnal variability but with a small longer-term trend during the event before increasing when the PCAP ends. Additionally, elevated $CO_2$ and depressed d-excess values were frequently observed in the absence of PCAPs (e.g., mid-December 2014 and 2016); these cases are associated with low mixing heights but not necessarily high VHD values, or of moderate VHD values that fell short of the VHD-based definition of a PCAP.

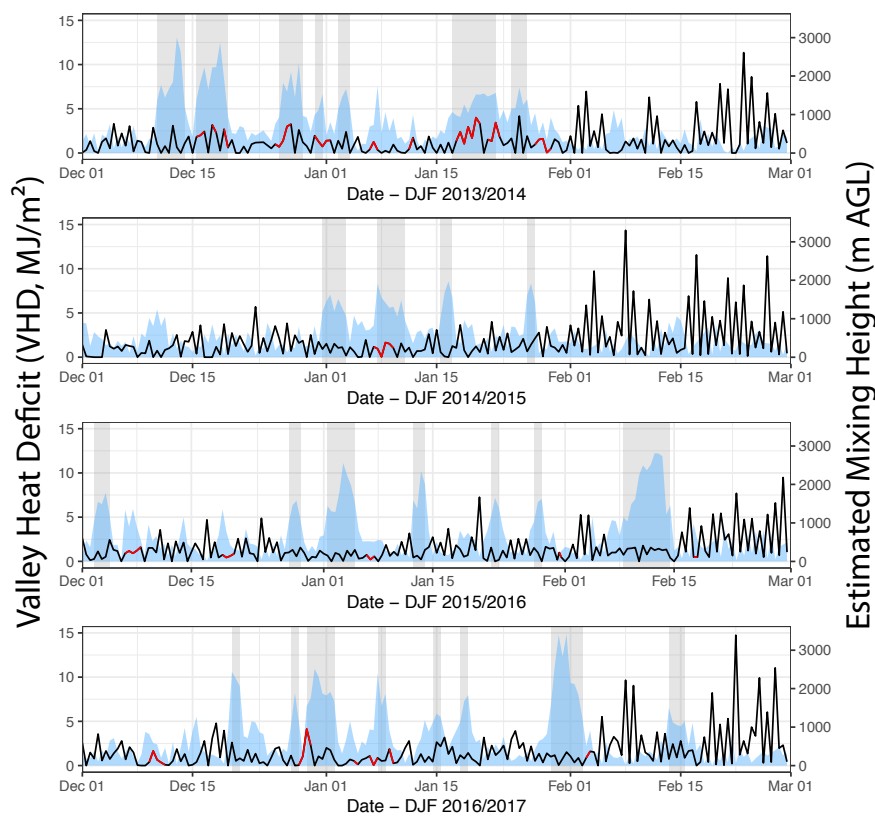

**Figure 2.** Valley heat deficit ($\mathrm{MJ\,m^{-2}}$, blue polygon) and mixing height (m, black indicates Richardson mixing height; red indicates surface-based inversion top) by season. Seven, four, seven, and eight PCAP events are identified for DJF 13/14, 14/15, 15/16, and 16/17, and are denoted by light gray shading.

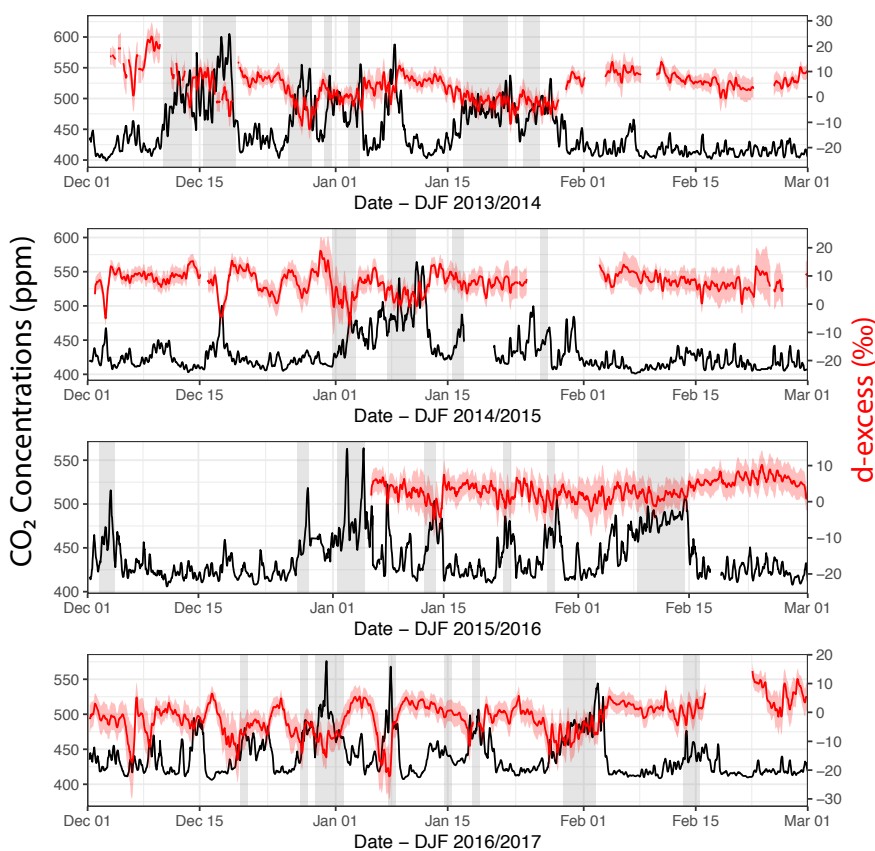

**Figure 3.** Six hour running-mean $CO_2$ concentrations (ppm, black line) and water vapor d-excess (‰ VSMOW, red line, $2\sigma$ uncertainty shown in red shading) measured at the UOU for DJF 2013-2017. Persistent cold air pool events are denoted by gray rectangles. When the lower atmosphere is stable, $CO_2$ builds up in the boundary layer and d-excess tends to decrease.

## 4.1 Relationship between $CO_2$ and d-excess and estimating d-excess of CDV

Clear distinctions emerged in the distributions of $CO_2$ and d-excess during PCAP events compared to more well-mixed periods. Non-PCAP periods are typically defined by lower $CO_2$ values, usually below $450\,\mathrm{ppm}$, and a broad range of d-excess values averaging around $\sim 10\%o$ and spanning $\sim 0-30\,\%o$ (Fig. 3). D-excess variability during non-PCAP periods is likely controlled by natural moistening and dehydration processes, including air mass mixing, Rayleigh-style condensation and evaporative inputs from the Great Salt Lake. In contrast, a strong linear relationship between $CO_2$ and d-excess is observed during PCAP periods, with d-excess values decreasing proportionally with increasing $CO_2$. At the highest $CO_2$ concentrations, d-excess can be >10%o lower than when $CO_2$ is at background levels outside of PCAP events.

These relationships between "natural" moistening and drying of the boundary layer and moistening by CDV become apparent from the relationship between d-excess and humidity (Fig. 4). We observe increasing $qd$ values with increasing $q$ at low $CO_2$ concentrations, but decreasing $qd$ values with increasing $CO_2$ (Fig. 4). Strong positive d-excess excursions are observed during the first two winters, and are associated with dry, cold conditions following the passage of a strong cold front. No equivalent excursions are observed during the last two winters, perhaps due to a similar magnitude cold front event not occurring during the observed portions of those winters. Negative excursions are observed during PCAP events or when $CO_2$ is elevated, and can be seen across a range of humidity values.

We leverage the observed, coupled variability in d-excess and $CO_2$ during periods of enhanced $CO_2$ to test previous theoretical estimates and limited direct measurements of $d_{\mathrm{CDV}}$ using a Keeling-style approach (1958; 1961). The best-fit slope of a linear mixed model allowing for random variation in the both the slope and intercept between PCAP events yields an estimate of $d_{\mathrm{CDV}}$ of $-179\pm17\%o$ for $ef = 1.5$ (Fig. 5). This estimate of $d_{\mathrm{CDV}}$ is consistent with the upper limit of theoretical estimates and pilot measurements from Gorski et al. (2015), and could be further validated by a comprehensive survey of fuels in the SLV. Based on this regression, d-excess decreases by $0.18\pm0.02\%o$ for every ppm increase in $CO_2$, though this rate of change will vary slightly with background $q$ (Fig. 1). Instrumental precision ($1\sigma$) for d-excess is estimated to be $2.4\,\%o$ at the mean DJF humidity value of $4\,\mathrm{mmol\,mol}^{-1}$, implying that enrichments of $\sim40$ ppm CDV can be detected at the $2\sigma$ level. This estimated detection limit will likely decrease as instrument precision and calibration routines are improved, and may change in other locations with different fuel use patterns and $ef$ values. For individual PCAPs, the slope of the regression and the strength of the correlation between $q_{\mathrm{obs}}d_{\mathrm{obs}}$ and $CO_2$ excess are more variable, with slopes ranging from $-25\pm43\%o$ to $-379\pm63\%o$ and coefficients of determination ranging from 0.77 to 0.001 (Table 3). The wide range of slopes and coefficients of determination observed hints at a complex relationship between urban humidity, $CO_2$, and CDV that varies with the nature of each period of high atmospheric stability. For example, fuel mixtures and heating demands may change with temperature, inversions based on the valley floor may trap most pollutants below the UOU observation site, and other sources and processes such as advection or evaporation over the Great Salt Lake may also contribute water vapor to the boundary layer and alter the relationship between $q_{\mathrm{obs}}d_{\mathrm{obs}}$ and $CO_2$ excess. Expanding observations beyond a single site (UOU) may help distinguish these possibilities.

Using this estimate for $d_{\mathrm{CDV}}$ of $-179\pm17\%o$, we estimate the maximum fraction of CDV for each PCAP event using equation 8 and estimates of $d_{\mathrm{bg}}$ from both the 12 hour period prior to PCAP initiation, or the last 12 hour period with a $CO_2$ minimum.

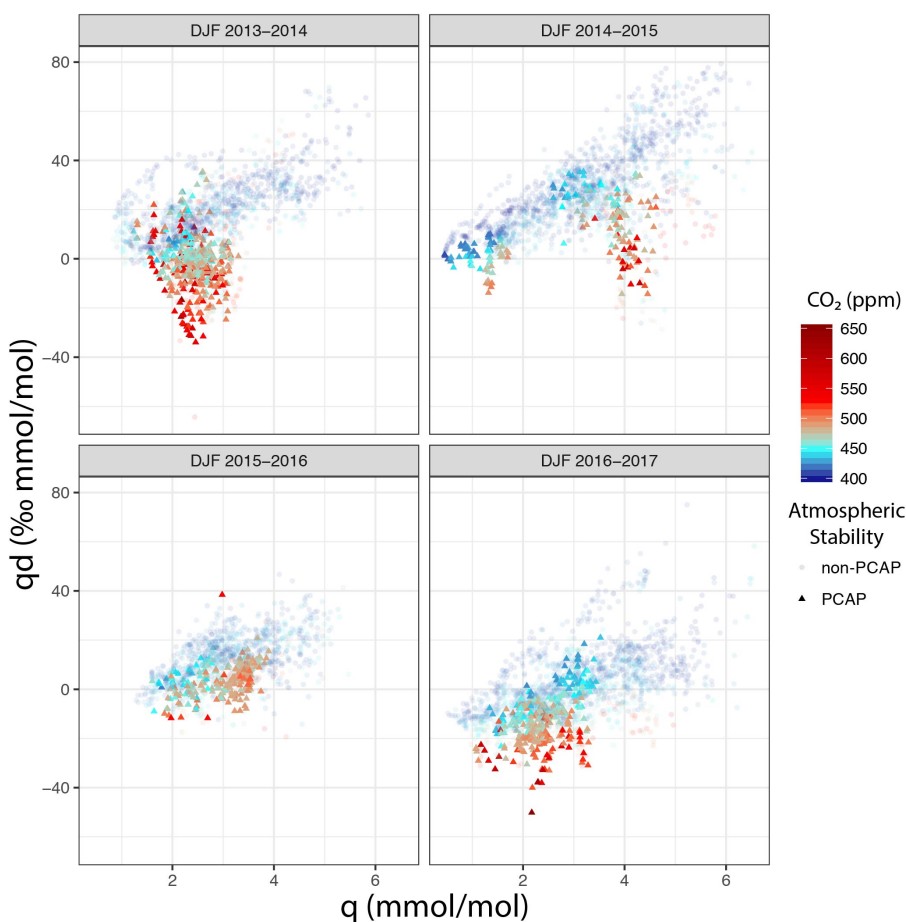

**Figure 4.** Relationship of the product of specific humidity and d-excess, $qd$ (‰ mmol mol$^{-1}$), against specific humidity $q$ (mmol mol$^{-1}$). Points are colored by $CO_2$ concentration (ppm) at the time of measurement, with the shape and opacity corresponding to whether the data point was collected during a PCAP event (opaque triangles) or outside of a PCAP event (semitransparent circles). Moistening and drying by condensation and mixing of "natural" air masses occurs along a line with positive slope, while moistening by CDV occurs along a line with negative slope.

When the former assumption is used for $d_{bg}$, estimates of the CDV fraction average 5.0% across all PCAP events, and range from $-2.1 \pm 2.3\%$ to $13.9 \pm 1.9\%$, while when the latter assumption for $d_{bg}$ is used, the mean CDV fraction rises to 7.2% and ranges from $2.2 \pm 2.1\%$ to $16.7 \pm 3.2\%$ (Table 4). Negative CDV fraction estimates occur when the estimated $d_{bg}$ is less than the minimum value of $d_{obs}$, and are only observed when the 12 hour period immediately preceding the initiation of the PCAP is used to estimate $d_{bg}$. $CO_2$ concentrations can build up whenever the atmosphere is stable, even if atmospheric stability has not yet met the PCAP threshold used here. Therefore, this pattern highlights the importance and challenge of accurately estimating $d_{bg}$ for this humidity apportionment method to yield accurate estimates of $q_{CDV}/q_{obs}$.

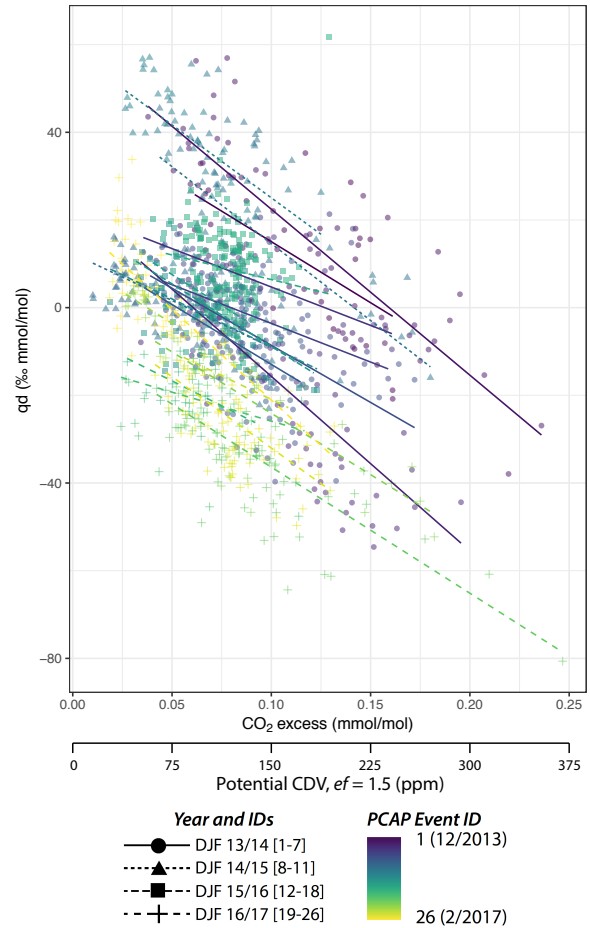

**Figure 5.** Miller-Tans style plots of $qd$ (‰ $\mathrm{mmol\,mol^{-1}}$) versus $CO_2$-excess (the difference between the observed $CO_2$ and the seasonal minimum $CO_2$) by year during PCAP events. The estimated d-excess of CDV, assuming CDV is the dominant flux of water into the boundary layer during PCAP events, is the slope of the best fit line.

## 4.2 Case studies

### 4.2.1 December 28, 2014-January 14, 2015

Two distinct PCAP events were observed between December 28, 2014 and January 14, 2015 (Fig. 6). The period prior to the first PCAP is marked by a cold front passage around December 30, 2014 12 UTC, where there are strong decreases in temperature and humidity (Fig. 6ab), elevated wind speeds (Fig. 6c), a $CO_2$ minimum (Fig. 6d), and an increase in d-excess to $\sim 18‰$ (Fig. 6) that is generally consistent with natural removal of water from the atmosphere (Fig. 6f). After onset of the PCAP, however, d-excess dropped rapidly as $CO_2$ and CDV began to build in the valley. By January 2, $CO_2$ had risen to 490 ppm and d-excess had fallen to $-7.4‰$, an increase of $\sim 60$ ppm and a decrease of $25‰$ respectively (Fig. 6de). Atmospheric

**Table 3.** Miller-Tans regression parameters for each PCAP event

| Start of PCAP | End of PCAP | Regression Slope ($ef = 1.5$) | Regression $R^2$ |
|---|---|---|---|
| 10 Dec 2013 1200 | 14 Dec 2013 0000 | $-190 \pm 46$ | 0.33 |
| 15 Dec 2013 1200 | 19 Dec 2013 1200 | $-260 \pm 21$ | 0.77 |
| 26 Dec 2013 0000 | 29 Dec 2013 0000 | $-275 \pm 27$ | 0.62 |
| 30 Dec 2013 1200 | 31 Dec 2013 1200 | $-89 \pm 45$ | 0.17 |
| 02 Jan 2014 1200 | 04 Jan 2014 0000 | $-101 \pm 41$ | 0.13 |
| 17 Jan 2014 0000 | 22 Jan 2014 1200 | $-173 \pm 25$ | 0.30 |
| 24 Jan 2014 1200 | 26 Jan 2014 1200 | $-185 \pm 35$ | 0.34 |
| 31 Dec 2014 1200 | 03 Jan 2015 1200 | $-134 \pm 22$ | 0.42 |
| 07 Jan 2015 1200 | 11 Jan 2015 0000 | $-241 \pm 39$ | 0.34 |
| 15 Jan 2015 1200 | 17 Jan 2015 0000 | $-228 \pm 46$ | 0.59 |
| 12 Jan 2016 1200 | 14 Jan 2016 0000 | $-128 \pm 38$ | 0.25 |
| 22 Jan 2016 1200 | 23 Jan 2016 1200 | $-199 \pm 39$ | 0.55 |
| 28 Jan 2016 0000 | 29 Jan 2016 0000 | $-206 \pm 99$ | 0.15 |
| 08 Feb 2016 1200 | 14 Feb 2016 1200 | $-25 \pm 43$ | 0.001 |
| 20 Dec 2016 0000 | 21 Dec 2016 0000 | $-130 \pm 54$ | 0.06 |
| 27 Dec 2016 1200 | 28 Dec 2016 1200 | $-45 \pm 54$ | 0.005 |
| 29 Dec 2016 1200 | 02 Jan 2017 0000 | $-193 \pm 18$ | 0.52 |
| 07 Jan 2017 1200 | 08 Jan 2017 1200 | $-189 \pm 39$ | 0.34 |
| 14 Jan 2017 1200 | 15 Jan 2017 1200 | $-379 \pm 63$ | 0.64 |
| 18 Jan 2017 0000 | 19 Jan 2017 0000 | $-41 \pm 30$ | 0.44 |
| 29 Jan 2017 1200 | 02 Feb 2017 1200 | $-232 \pm 32$ | 0.08 |
| 13 Feb 2017 1200 | 15 Feb 2017 1200 | $-328 \pm 40$ | 0.62 |

d-excess through this period closely followed model expectations of moistening via CDV (Fig. 6f). After the end of the first PCAP event, specific humidity and temperature rose daily until the start of the second PCAP on January 7, 12 UTC (Fig. 6ab). During this period in between PCAP events, $CO_2$ remained elevated, and exhibited diurnal variability of 20-40 ppm (Fig. 6d), but d-excess remained more consistent (Fig. 6e). Together, the pattern of d-excess and $CO_2$ change across between the two PCAP events is consistent with "natural" moistening of the boundary layer paired with an incomplete mix-out of CDV and $CO_2$. The second PCAP event, spanning January 7 12 UTC until January 11 00 UTC, was marked by prominent diurnal cycles in humidity, temperature, and $CO_2$ (Fig. 6a,b,d). Strong diurnal cyclicity was also observed in d-excess (Fig. 6e). $CO_2$ concentrations reached their maximum at the end of the PCAP event, and decreased slowly during the first diurnal cycle after the breakup of the PCAP, before mixing out nearly completely on January 12. D-excess values followed changes in $CO_2$,

**Table 4.** Estimates of CDV humidity fraction

| Start of PCAP | End of PCAP | Min $d_{obs}$ | Estimated $d_{nat}$ (12h mean before PCAP) | Estimated $d_{nat}$ (last 12h period with $CO_2 < 415$ ppm) | $q_{CDV}/q_{obs}$ (12h mean before PCAP) | $q_{CDV}/q_{obs}$ (last 12h period with $CO_2 < 415$ ppm) |
|---|---|---|---|---|---|---|
| 10 Dec 2013 1200 | 14 Dec 2013 0000 | $-7.0 \pm 2.3$ | $20.8 \pm 0.5$ | $20.3 \pm 1.7$ | $13.9 \pm 1.9$ | $13.7 \pm 2.4$ |
| 15 Dec 2013 1200 | 19 Dec 2013 1200 | $-10.9 \pm 2.0$ | $7.7 \pm 1.2$ | $7.5 \pm 1.4^c$ | $10.0 \pm 2.0$ | $9.9 \pm 2.1$ |
| 26 Dec 2013 0000 | 29 Dec 2013 0000 | $-13.8 \pm 1.9$ | $2.6 \pm 1.5$ | $7.0 \pm 1.4$ | $9.0 \pm 2.1$ | $11.2 \pm 2.1$ |
| 30 Dec 2013 1200 | 31 Dec 2013 1200 | $-4.1 \pm 1.8$ | $4.9 \pm 1.4$ | $0.6 \pm 1.4^b$ | $4.9 \pm 1.8$ | $2.6 \pm 1.8$ |
| 02 Jan 2014 1200 | 04 Jan 2014 0000 | $-8.1 \pm 1.6$ | $0.3 \pm 1.3$ | $0.7 \pm 1.3^c$ | $4.7 \pm 1.7$ | $4.9 \pm 1.7$ |
| 17 Jan 2014 0000 | 22 Jan 2014 1200 | $-9.6 \pm 1.8$ | $-1.0 \pm 1.4$ | $8.3 \pm 1.3$ | $4.8 \pm 1.9$ | $9.6 \pm 1.9$ |
| 24 Jan 2014 1200 | 26 Jan 2014 1200 | $-7.8 \pm 2.2$ | $1.3 \pm 1.4$ | $1.8 \pm 1.4^b$ | $5.0 \pm 2.1$ | $5.3 \pm 2.1$ |
| 31 Dec 2014 1200 | 03 Jan 2015 1200 | $-10.5 \pm 2.6$ | $9.7 \pm 2.2^d$ | $9.7 \pm 2.2^d$ | $10.7 \pm 2.8$ | $10.7 \pm 2.8$ |
| 07 Jan 2015 1200 | 11 Jan 2015 0000 | $-3.6 \pm 1.3$ | $3.5 \pm 1.2$ | $12.6 \pm 1.3^b$ | $3.9 \pm 1.4$ | $8.5 \pm 1.6$ |
| 15 Jan 2015 1200 | 17 Jan 2015 0000 | $2.2 \pm 2.0$ | $10.8 \pm 1.4$ | $8.4 \pm 1.4^b$ | $4.5 \pm 1.8$ | $3.3 \pm 1.8$ |
| 12 Jan 2016 1200 | 14 Jan 2016 0000 | $-5.9 \pm 2.2$ | $2.6 \pm 1.7$ | $3.2 \pm 1.7$ | $4.7 \pm 2.2$ | $5.0 \pm 2.2$ |
| 22 Jan 2016 1200 | 23 Jan 2016 1200 | $-4.3 \pm 1.9$ | $2.0 \pm 3.6$ | $3.6 \pm 1.6$ | $3.5 \pm 2.0$ | $4.3 \pm 2.0$ |
| 28 Jan 2016 0000 | 29 Jan 2016 0000 | $-3.4 \pm 2.1$ | $-1.1 \pm 1.5$ | $2.0 \pm 1.6^b$ | $1.3 \pm 2.0$ | $3.0 \pm 2.1$ |
| 08 Feb 2016 1200 | 14 Feb 2016 1200 | $-2.7 \pm 1.9$ | $2.2 \pm 1.4$ | $2.6 \pm 1.3$ | $2.7 \pm 1.8$ | $2.9 \pm 1.8$ |
| 20 Dec 2016 0000 | 21 Dec 2016 0000 | $-9.8 \pm 2.3$ | $-12.9 \pm 2.0$ | $2.5 \pm 1.3$ | $-1.9 \pm 2.8$ | $6.8 \pm 2.3$ |
| 27 Dec 2016 1200 | 28 Dec 2016 1200 | $-17.0 \pm 2.9$ | $-8.4 \pm 1.7$ | $-3.5 \pm 1.4$ | $5.0 \pm 2.8$ | $7.7 \pm 2.6$ |
| 29 Dec 2016 1200 | 02 Jan 2017 0000 | $-23.1 \pm 2.3$ | $-7.6 \pm 1.5$ | $-6.0 \pm 1.4^c$ | $9.0 \pm 2.4$ | $9.9 \pm 2.4$ |
| 07 Jan 2017 1200 | 08 Jan 2017 1200 | $-25.9 \pm 3.9$ | $-18.0 \pm 2.0$ | $4.7 \pm 1.2$ | $4.9 \pm 3.7$ | $16.7 \pm 3.2$ |
| 14 Jan 2017 1200 | 15 Jan 2017 1200 | $-2.4 \pm 1.9$ | $0.6 \pm 1.4$ | $4.5 \pm 1.2$ | $1.7 \pm 1.8$ | $3.8 \pm 1.7$ |
| 18 Jan 2017 0000 | 19 Jan 2017 0000 | $-4.9 \pm 2.3$ | $-8.4 \pm 1.6$ | $-0.9 \pm 1.4^b$ | $-2.1 \pm 2.3$ | $2.2 \pm 2.1$ |
| 29 Jan 2017 1200 | 02 Feb 2017 1200 | $-14.7 \pm 3.1$ | $-7.8 \pm 1.7$ | $3.8 \pm 1.3$ | $4.0 \pm 2.8$ | $10.1 \pm 2.6$ |
| 13 Feb 2017 1200 | 15 Feb 2017 1200 | $-9.4 \pm 2.1$ | $1.0 \pm 1.4$ | $1.2 \pm 1.2$ | $5.8 \pm 2.0$ | $5.9 \pm 1.9$ |

a: $d_{bg}$ estimated with 415 ppm < $CO_2$ < 425 ppm

b: $d_{bg}$ estimated with 425 ppm < $CO_2$ < 450 ppm

c: $d_{bg}$ estimated with 450 ppm < $CO_2$ < 475 ppm

d: both $d_{bg}$ estimates from the same observation

remaining low but increasing with decreasing $CO_2$ during the first diurnal cycle, before rapidly increasing as $CO_2$ decreased at the end of the observation period (Fig. 6e). The spike in $CO_2$ at the end of the PCAP is likely due to the UOU's location on a topographic bench; strong stability during the PCAP may have kept the most polluted air below the UOU, which then was transported to higher altitudes as the PCAP ended.

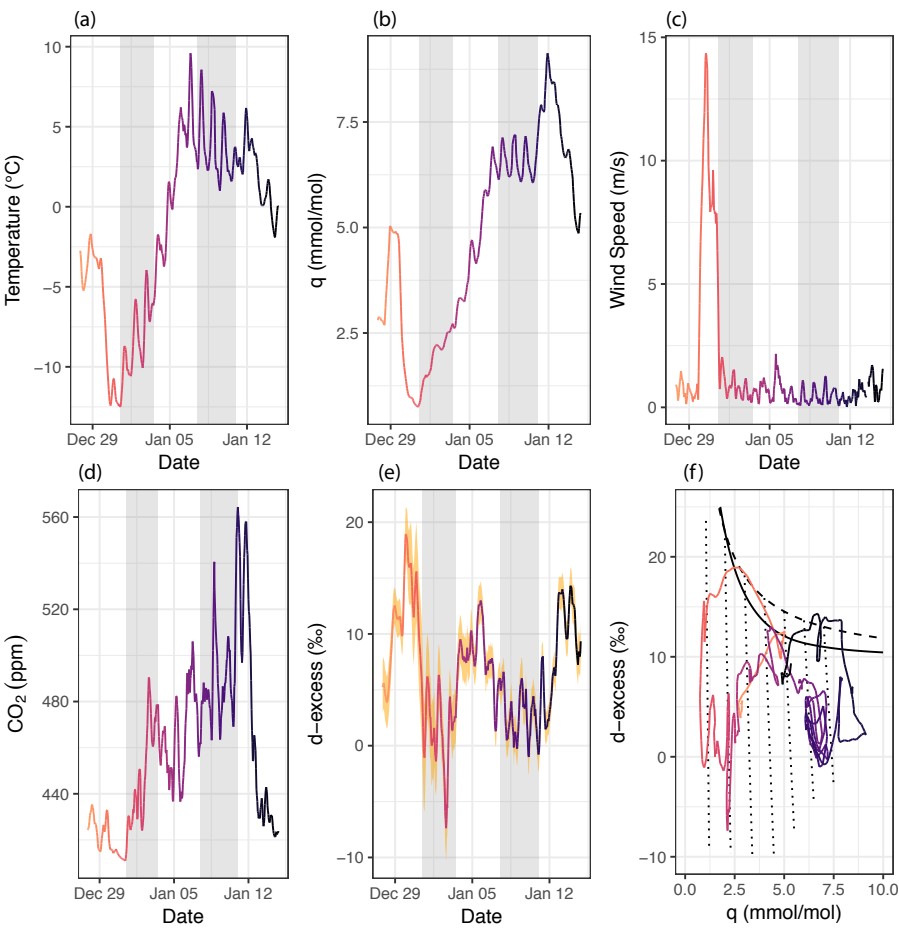

**Figure 6.** Relationships between meteorology, d-excess, and $CO_2$ from December 28, 2014-January 14, 2015. Time series of temperature (a, °C), $q$, (b, mmol mol$^{-1}$), wind speed (c, m s$^{-1}$), $CO_2$ (d, ppm; $1\sigma$ uncertainty in orange shading), d-excess (e, ‰), and the relationship between $dq$ vs $q$ (f) spanning the same time period, with the same color gradient used across time in all four panels. Data are plotted as 6-hour running averages.

### 4.2.2 February 3-17, 2016

This period was marked by one extended PCAP from February 8, 12 UTC to February 14, 12 UTC (Fig. 7), and has been a major focus of recent air pollution studies (Baasandorj et al., 2017; Bares et al., 2018). Conditions prior to the PCAP were dry and cold for the first two days, before warming by $\sim 5°$C (Fig. 7a), concurrent with an increase in humidity from $\sim 3$ to $\sim 5\,\mathrm{mmol\,mol^{-1}}$ (Fig. 7b). Wind speeds peaked at the beginning of this period, and remained below $2\,\mathrm{m\,s^{-1}}$ after February 5 (Fig. 7c). $CO_2$ increased from 430 to 480 ppm before decreasing back to 430 ppm (Fig. 7d). Deuterium excess also decreased by a few permil while $CO_2$ was elevated, but increased back to 3-5‰ until the beginning of the PCAP (Fig. 7e); humidity increased rapidly during this period, and followed a path parallel to moistening by addition of "natural" water vapor (Fig. 7f). The remainder of the pre-PCAP period through the PCAP event was marked by slow, steady increases in $q$ and $CO_2$, with prominent diurnal cycling in temperature, $CO_2$, $q$, and d-excess. Diurnal cyclicity was apparent in the relationship between d-excess and $CO_2$ as well, with periods of increasing (decreasing) $CO_2$ producing rapid decreases (increases) in d-excess with little change in $q$. These diurnal patterns are consistent with daytime growth of a shallow convective boundary layer at the surface with a stable layer aloft; the same interpretation was made in prior studies of this event (Baasandorj et al., 2017). Diurnal cycle amplitudes of $q$, temperature, and $CO_2$ decreased for the second half of the PCAP (Fig. 7a,b,d), and co-occur with a reduction in surface solar radiation as low-level clouds developed during the event. Superimposed on these diurnal cycles of d-excess against $q$, conditions became more moist across several days (Fig. 7b,f). Following termination of the PCAP, conditions became warmer and $CO_2$ decreased back toward its background value. Humidity increased rapidly for a few days after the event before falling again. Both the moistening and drying occurred with small changes in d-excess, consistent with changes expected for changes in $q$ in the absence of the buildup of CDV. In contrast to the previous case study, the relationship between d-excess and $CO_2$ excess is weak across this PCAP event (Table 3). Atmospheric soundings indicate the presence of a shallow convective mixed layer near the surface topped by a strong temperature inversion during this event (e.g., Baasandorj et al., 2017), suggesting that the column within which $CO_2$ and CDV are emitted may larger than for PCAPs with high atmospheric stability lower in the column. Although changes in $q$ across multiple days during this event seem to be driven by processes other than CDV addition, these observations support a strong CDV contribution on diurnal timescales as d-excess values and $CO_2$ concentrations are correlated at diurnal timescales but not necessarily multi-day timescales during this event.

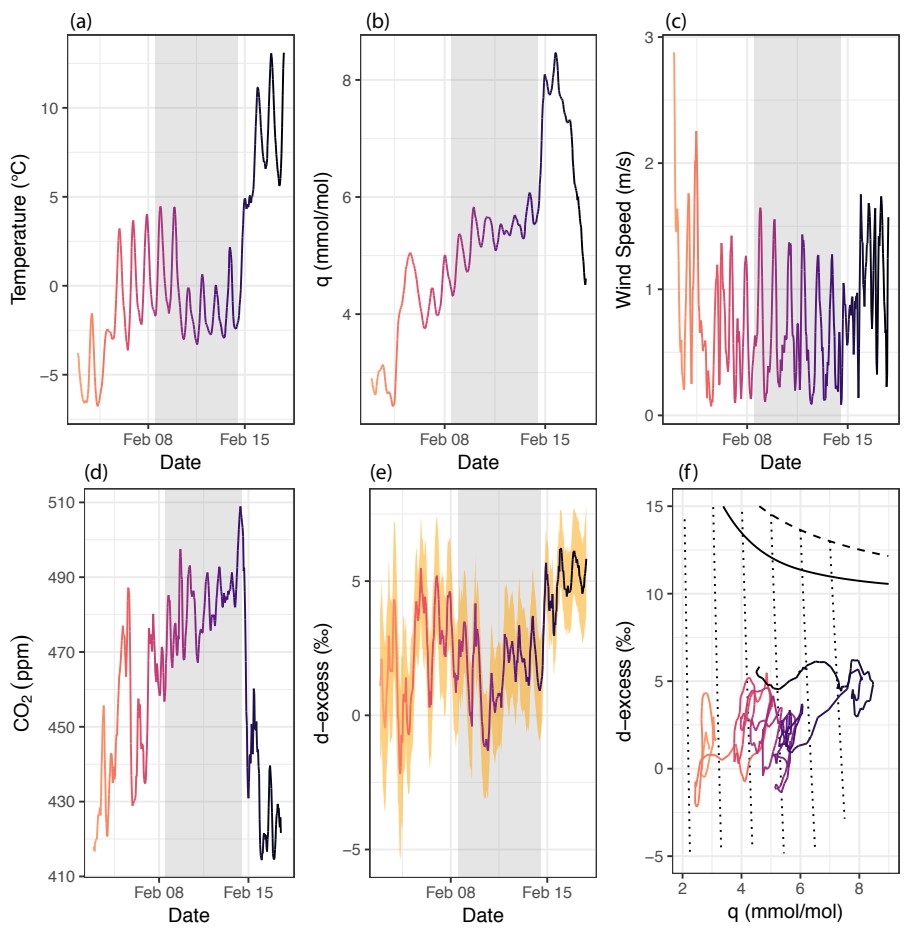

**Figure 7.** Relationships between meteorology, d-excess, and $CO_2$ from February 3-17, 2016. Time series of temperature (a, °C), $q$, (b, $mmol\,mol^{-1}$), wind speed (c, $m\,s^{-1}$), $CO_2$ (d, ppm; $1\sigma$ uncertainty in orange shading), d-excess (e, ‰), and the relationship between $dq$ vs $q$ (f) spanning the same time period, with the same color gradient used across time in all four panels. Data are plotted as 6-hour running averages.

## 4.3 Diurnal cycles of humidity, $CO_2$, and d-excess

In this section, we more closely examine diurnal cycles of d-excess, $CO_2$, and specific humidity. We define diurnal cycles as deviations from the 24-hour running mean, and indicate them with a capital delta ($\Delta$). Changes in the diurnal variability of the estimated mixing height and valley heat deficit were apparent throughout the winter season (Fig. 2). Despite subtle variation of the diurnal cycles of $\Delta$d-excess, $\Delta CO_2$, and $\Delta q$ across years and months, several robust patterns emerged (Fig. 8). $\Delta$d-excess was flat or increased slightly in the early morning hours (0-6 Local Time, LT), decreased throughout the morning until $\sim$11 LT, increased from 11 LT until late afternoon ($\sim$17 LT), and then decreased again from 17 LT until late evening (Fig. 8a,d). The mean amplitude of the $\Delta$d-excess diurnal cycle was $\sim 6\%o$ during PCAP events (Fig. 8a), and closer to $\sim 3\%o$ during non-PCAP periods (Fig. 8d).

Daily minimums in $CO_2$ mirror daily maximums in d-excess, and occurred during the the afternoon, when convective mixing, and therefore exchange between the surface and air aloft, is greatest (Fig. 8b,e). Conversely, daily minimums in $\Delta$d-excess occur when $\Delta CO_2$ is increasing, likely reflecting the addition of CDV. Like $\Delta$d-excess, the amplitude of the diurnal cycle for $\Delta CO_2$ is greater during PCAP periods ($\sim 40$ ppm, Fig. 8b) than during non-PCAP periods ($\sim 20$ ppm, Fig. 8e). Patterns in $\Delta$d-excess diurnal cycles mirrored $\Delta CO_2$ patterns, demonstrating the close association between d-excess and $CO_2$ on short time scales. In contrast, diurnal cycles of $\Delta q$ show different patterns apart from amplitude across PCAP and non-PCAP periods (Fig. 8c,f). During PCAP periods, $\Delta q$ increases from $\sim 6$ LT to $\sim 18$ LT, and decreases from $\sim 18$ LT to $\sim 6$ LT (Fig. 8c), with an amplitude of 0.7-0.8 $\mathrm{mmol\,mol^{-1}}$ across the day. During non-PCAP periods, the amplitude of the $\Delta q$ diurnal cycle decreased to $\sim 0.4\ \mathrm{mmol\,mol^{-1}}$, and features a period stable humidity or slight humidity decrease during the afternoon, presumably due to greater mixing between the boundary layer and the free troposphere (Fig. 8f). Interannual variability in the diurnal cycles was generally small, with the largest differences observed during PCAP periods. For example, composite diurnal cycles for PCAP events varied the most across years (Fig. 8a-c). However, given the episodic nature of PCAPs, these diurnal cycles can often be determined by 1 or 2 events in a given year. Though a consistent pattern emerged across many PCAP events, individual events were expressed differently in both the $CO_2$ and d-excess records (e.g., section 4.2). Nonetheless, the close associations between d-excess and $CO_2$ on diurnal cycles, coupled with the observation that these cycles are generally not coherent with changes in specific humidity, further suggest that the observed d-excess variability reflects the addition or removal of CDV.

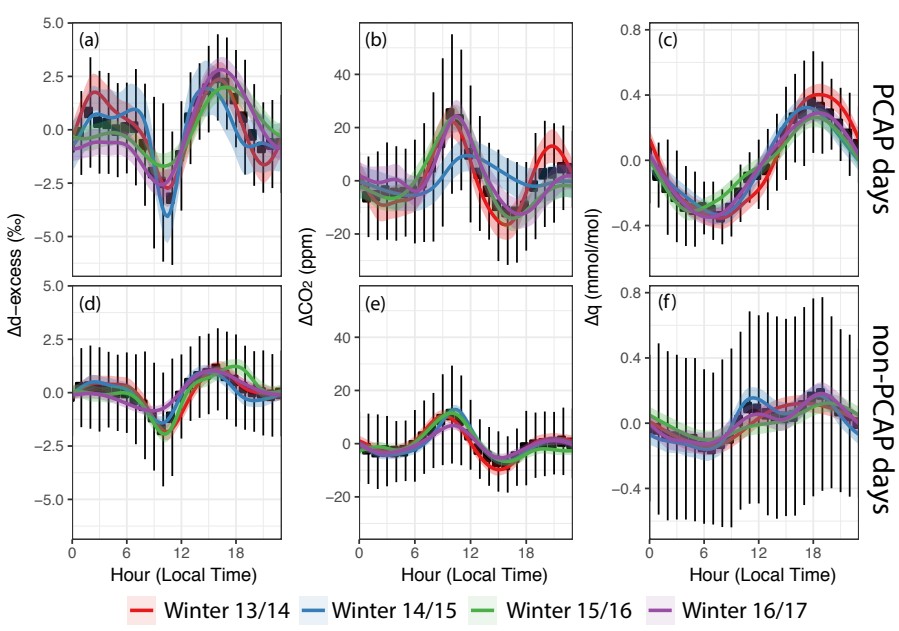

**Figure 8.** Seasonal average diurnal cycles of $\Delta$d-excess (left column), $\Delta CO_2$ (center column), and $\Delta q$ (right column) for days in PCAP conditions (top row) or non-PCAP conditions (bottom row). The diurnal cycle is approximated here as the deviation from a 24-hr moving average. Mean values across all four years are shown as black symbols, with black vertical lines indicating $1\sigma$ variability. The mean diurnal cycle is modeled for each year independently as a GAM using cubic cyclic smoothing splines, and regression standard error shown as shaded ribbons, with the color corresponding to model year.

## 5 Discussion

CDV is evident across sub-diurnal to multi-day timescales in the Salt Lake City d-excess record. On short timescales, periods of high emission intensity were apparent in the diurnal cycles of d-excess and $CO_2$. Decreases in d-excess were coincident with increases in $CO_2$, and occur during the morning and late afternoon when emissions were likely high and tropospheric
mixing was low. Average diurnal cycles in d-excess and $CO_2$ showed little change overnight outside of PCAP events (Fig. 8), which was unexpected as heating emissions continued throughout the evening. The absence of overnight d-excess and $CO_2$ changes was likely a result of the UOU's location on a topographic bench away from large residential areas, or due to injection of cleaner air from above if a surface-based inversion occurs at an elevation below the UOU site. Long-term records of $CO_2$ have also been collected in lower-elevation areas of the SLV and exhibit a greater buildup of $CO_2$ overnight during the winter
than observed at UOU (Mitchell et al., 2018), which suggests that a stronger trend in nighttime d-excess and $CO_2$ values might be observed elsewhere in the SLV.

On longer timescales, the impact of CDV was most apparent during PCAP events, where $CO_2$ and CDV persist in the urban atmosphere while the atmosphere in the SLV remained sufficiently stable. Some contrasts in the expression of CDV and $CO_2$ were apparent across the winter season and likely resulted from changes in insolation and the mechanisms resulting in stability
of the near-surface atmosphere. For example, the most rapid increases in $CO_2$ and decreases in d-excess were observed during December and January (Fig. 3, 6), when surface insolation was lower. In contrast, a strong diurnal cycle but a more muted multi-day response was observed in February, when higher insolation can drive higher mixing heights (Fig. 2) and mix out a greater proportion of daily emissions. As a result, changes in d-excess and $CO_2$ exhibited large diurnal cycles superimposed upon slower synoptic trends during February PCAP events (Fig. 7).

Based on changes in d-excess relative to $CO_2$ during PCAP events, and the HESTIA inventory of fossil fuel emissions for SLV (Patarasuk et al., 2016), we have estimated the mean d-excess of CDV to be $-179 \pm 17$‰. One assumption of the model used here is that all of the change in d-excess is driven by addition of CDV; other sources of vapor to the near surface, such as sublimation of snow or water evaporated from the Great Salt Lake, may introduce bias into these estimates. However, both of these sources would have less negative d-excess values, and therefore, if other sources of vapor contribute significantly to
d-excess change, our estimates of $d_{CDV}$ are a maximum estimate. Deposition of vapor onto ice in supersaturated conditions can also promote a decrease in vapor d-excess (Galewsky et al., 2011; Jouzel and Merlivat, 1984). While we do not have any direct observations of supersaturated conditions, we cannot rule out the possibility of supersaturated conditions occurring when snow is in the valley or during cloud formation. However, we expect any potential role for vapor deposition under supersaturated conditions affecting vapor d-excess to be small, as we do not typically observe decreases in d-excess concurrent with decreases
in specific humidity (Fig. 8).

We have made an estimate of 1.5 for $ef$ through a detailed accounting of emissions or fuel sources from the HESTIA dataset (e.g., Patarasuk et al., 2016), but several sources of uncertainty in net $ef$ remain. For example, heat exchangers designed to improve heating efficiency may reduce the $H_2O$ concentration in emissions, and potentially alter $d_{CDV}$ as well through condensation of water in the emissions stream (Fig. 1). Additionally, the portfolio of fuels contributing to CDV change in

both time and space, and respond to meteorological conditions. For example, colder conditions increase demand for heating, which may shift the portfolio of fuel sources toward natural gas (e.g., Pataki et al., 2006). Finally, $d_{CDV}$ can be altered by the temperature and degree of equilibration of $^{18}O$ between $H_2O$ and $CO_2$ in combustion exhaust. If no equilibration occurs between $H_2O$ and $CO_2$, the $\delta^{18}O$ values of both species should be equal to atmospheric oxygen, 23.9‰ (Barkan and Luz,

2005; Gorski et al., 2015). In contrast, equilibration between $H_2O$ and $CO_2$ will lower the $\delta^{18}O$ value of $H_2O$; at $100°$, for example, the $\delta^{18}O$ value of $H_2O$ will be $\sim 29‰$ lower than the $\delta^{18}O$ of $CO_2$ for complete equilibration (Friedman and O'Neil, 1977; Gorski et al., 2015). The degree of equilibration may vary across fuels and combustion systems (Horváth et al., 2012), which introduces uncertainty into the $\delta^{18}O$, and subsequently $d$, of CDV. Regardless, the highly negative estimated isotopic composition of the flux into the boundary layer during PCAP events, which we have assumed is predominantly CDV, precludes

other potential sources of water vapor apart from CDV from explaining the observed isotopic change. These methods may also be helpful to verify that background $CO_2$ measurements are free from local emissions, as we would not expect to see a strong correlation between $CO_2$ concentrations and d-excess values in the absence of local emissions.

Though the most prominent periods of $CO_2$ and CDV buildup occur during PCAP events, decreases in d-excess coincident with increases in $CO_2$ were apparent outside of PCAPs as well. $CO_2$ and CDV from emissions built up in the boundary layer

whenever atmospheric stability was present regardless of whether VHD values were high enough to qualify as a PCAP. For a given quantity of fuel burned, $CO_2$ increases and CDV concentrations will be higher if the mixed height is lower because the volume these species mix into is smaller. Atmospheric soundings at the Salt Lake City airport occurred at 5 and 17 LT, and were unlikely to capture diurnal extremes in the mixing height, confounding efforts to develop high-frequency relationships between mixing height, $CO_2$, and CDV. Mid-afternoon patterns in the diurnal cycles of d-excess and $CO_2$ suggested that boundary

layer development and entrainment did mix a fraction of combustion products out of the boundary layer. This pattern held even during PCAP events (Fig. 8a,b), though it is not clear whether this reflects mixing out of the valley, or just a repartitioning of pollutants within the atmospheric column below a capping inversion. In contrast, $CO_2$ and CDV build to higher concentrations during the early morning and late afternoon (Fig. 8), when boundary layer mixing was decreased and emissions were likely higher due to elevated traffic.

This technique for measuring water from combustion in urban areas can be adapted beyond the SLV, though different environments will present distinct challenges. The SLV is well-suited to detecting the buildup of CDV as it has a dry climate, features a large urban area in a topographic basin, and experiences frequent multi-day periods of high atmospheric stability in the winter. The CDV signal is largest in dry regions or during winter (Fig. 1), and CDV may comprise a larger fraction of urban humidity in these cities for a given level of emissions intensity. Additionally, CDV may have a larger impact on the

radiative balance of cities in drier regions, as longwave forcing increases logarithmically with water vapor amount (Raval and Ramanathan, 1989). However, though the CDV signal is higher at low humidities, instrumental precision is lower. Therefore, at current instrumental precision limits, there is a trade-off between precision of the CDV estimates and the size of the CDV signal. Based on our study, we suggest two potential refinements to this technique that will improve the accuracy and precision of this technique to diagnose the fraction of urban humidity arising from CDV. First, the largest source of known uncertainty in

our estimates is associated with $d_{CDV}$. While our estimate of $-179 \pm 17‰$ is consistent with theoretical estimates, this fraction

may vary through time as a result of changing fuel mixtures (affecting both isotopic composition and $ef$) or measurement footprints, and has not been rigorously validated with direct measurements of $d_{CDV}$ from a wide variety of fuel sources and combustion systems. Additionally, due to spatial variability in the $\delta^2H$ composition of fuels, $d_{CDV}$ likely varies for other cities. Second, the estimate of the urban CDV fraction of humidity is highly sensitive to the estimate of $d_{bg}$. In this study, estimates of the CDV humidity percentage were 2.2% greater on average when a low $CO_2$ threshold was used rather than one based on the time window immediately preceding the PCAP; in one case, these assumptions yielded estimates that varied by a factor of 3.4, and in other cases, even yielded different signs (Table 4). In our uncertainty analysis, we have considered uncertainty arising from instrumental precision, but the uncertainty in $d_{bg}$ remains difficult to assess. Paired urban-rural observations may be necessary to accurately estimate $d_{bg}$, or identify appropriate periods for estimating $d_{bg}$ from the urban record.

## 6  Conclusions

Measurements of ambient vapor d-excess were paired with $CO_2$ observations across four winters in Salt Lake City, UT. We found a strong negative association between $CO_2$ and d-excess on sub-diurnal to seasonal timescales. Elevated $CO_2$ and CDV was most prominent during PCAP periods, where atmospheric stability was high for extended periods. We outline theoretical models that can discriminate between changes in d-excess driven by condensation, advection, and mixing processes the "natural" hydrological cycle and those driven by CDV moistening. The CDV signal is largest when humidity is low, as CDV likely comprises a larger fraction of total humidity and the anticipated signal between vapor with and without CDV is large. On shorter timescales, prominent diurnal cycles were observed in both d-excess and CDV that could be tied to both emissions intensity and atmospheric processes. These diurnal cycles were decoupled from diurnal cycles of specific humidity, further strengthening the link between d-excess and urban $CO_2$.

We estimate the d-excess value of CDV to be $-179 \pm 17\text{‰}$ assuming a mean molar ratio of $H_2O$:$CO_2$ in emissions of 1.5 derived from the HESTIA inventory of emissions for Salt Lake County (Patarasuk et al., 2016; Gurney et al., 2012). This estimate is consistent with theoretical constraints and a limited number of direct observations of CDV (Gorski et al., 2015), though uncertainty remains due to variability in the valley-scale stoichiometric ratio of $H_2O$ and $CO_2$ and the measurement footprint, and uncertainties about the isotopic composition of fuels and their transit through different combustion systems. The latter of these uncertainties can be reduced in future studies that seek to generate a "bottom-up" estimate of $d_{CDV}$ from direct measurements of fuels and emissions vapor to complement the "top-down" estimate made in this study using a mixing-model approach. We use our $d_{CDV}$ estimate to calculate the fraction of humidity in the SLV comprised of CDV using two different assumptions for the d-excess of water vapor in the absence of fossil fuel emissions. We find that CDV generally represents 5-10% of urban humidity during PCAP events, with a maximum estimate of $16.7 \pm 3.2\%$. Estimates of urban CDV fraction require an accurate estimate of the d-excess of water vapor in the absence of emissions, and we find generally higher estimates of urban CDV when a low-$CO_2$ threshold is used to estimate $d_{bg}$ compared to when pre-PCAP observations alone are used. Further refinements of these methods may help apportion humidity changes during the winter between CDV and different advected "natural" water sources to the urban environment, and help verify that $CO_2$ measurements that are taken

as backgrounds are not influenced by local emissions. Additionally, our method is most immediately applicable to cities in arid or semi-arid areas during the winter, as the potential isotopic signal for detecting CDV is the largest. However, CDV may have the largest impact on urban meteorology when humidity is low, as greenhouse forcing by water vapor is logarithmically proportional to water vapor concentration. Further refinements of this humidity apportionment technique, such as narrowing

5    the uncertainty in the isotopic composition of CDV and improving the estimation of $d_{bg}$ will improve estimates of CDV amount in urban environments, and help assess relationships between CDV, $CO_2$, urban air pollution, and public health.

*Code and data availability.*   IGRA radiosonde data are available from https://www.ncdc.noaa.gov/data-access/weather-balloon/integrated-global-radioson UOU meteorological measurements are available for download from mesowest.utah.edu, and $CO_2$ data are available at air.utah.edu. Calibrated UOU isotope data products are available from the Open Science Framework (osf.io/ekty3), and codes used to calibrate the water

10    isotope analyzer measurements are available from GitHub (https://github.com/SPATIAL-Lab/UU_vapor_processing_scripts/releases/tag/v1. 2.0).

*Competing interests.*   The authors declare that they have no conflicts of interest.

*Acknowledgements.*   RPF and GJB received support from NSF grant EF-1241286. RB, JCL, and the $CO_2$ measurements were supported by grants from Department of Energy (DOE) grant DESC0010624 and the National Oceanic and Atmospheric Administration (NOAA) grant

15    NA140AR4310178.

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
