# Peer review of "Detection and variability of combustion-derived vapor in an urban basin"

_Atmospheric Chemistry and Physics, 2017_

## Referee Comment (RC1) · Anonymous Referee #1 · 24 Feb 2018

General Comments:

This study evaluates a Keeling-style approach for determining the deuterium-excess signature of combustion derived water vapor (CDV) in the Salt Lake City area. The new approach is consistent with values reported in the group's earlier paper, Gorski et al., 2015. The paper also develops criteria for filtering observational periods when atmospheric conditions are most conducive to the accumulation of CDV. These criteria could be used as a starting point for similar studies conducted in other cities. This is the first study that reports multiple years of water vapor isotope measurements to study CDV. While the study is certainly novel, and the quantification of CDV is important, I think the authors could improve the paper by (1) explicitly stating why this study is important using detailed examples, (2) discussing the broader impacts of the work (how does this work further the field, and where else are improvements needed), and (3) providing quantitative support to put the results of the study into context. For example, the reasons some parameter values are used (e.g. emission factors (ef) from 1-2, CDV mole fractions ranging from 100-500 ppm) should be supported with more explanation. The paper is well-written and concise. The specific suggestions listed below, if incorporated, will provide readers with greater context for interpreting the results of the study.

Specific Comments:
1. Pg. 1. Ln 21. This might be the only sentence in the paper that explicitly states why quantifying CDV emissions is important. Could you expand this idea by detailing possible CDV impacts in urban areas, e.g. impacts on downwind clouds/weather, link between enhanced humidity/temperatures and heat stroke/fatalities in at-risk groups (elderly, sick), influence on photochemistry/aerosol, etc.
2. Pg. 2. Ln 11. This is an appropriate place to introduce the idea of the SLV's seasonally shifting fossil fuel use (and $H_2O:CO_2$ combustion stoichiometry), which adds to the complexity of quantifying CDV emissions. Furthermore, fossil fuel use trends differ from city to city. Describing the complexities of (1) CDV isotope measurements and (2) uncertainties regarding stoichiometry, fossil fuel consumption, and the impacts on CDV d-excess and emissions estimates bolsters your statements regarding the need for refinements to the method (last sentence of abstract). It would also help to communicate the novelty of these types of studies, and the need to continue work in this area.
3. Pg. 2. Ln 16-19. The last line of the introduction indicates that an objective of this study is to investigate relationships involving CDV amount. Does your analysis allow you to report CDV contributions to the SLC boundary layer (Gorski et al reports up to 13% CDV), or do you mean to say your approach allows for the estimation of CDV mole fractions (based on CDV moistening lines in Figure 1, 7-9), or do you mean to say this study intends to report general relationships between atmospheric stability and CDV amount (not necessarily quantitative estimates). Please clarify.
4. Pg 3 ln 4. Why is 2200 msl used in the VHD equation? Is it because that's roughly the height of the mountains surrounding the SLV? Or does it have to do with average mixing height (1290 m + 1500 m = 2790 m, so maybe not?)

5. Pg 3. Ln 6. You reference Whiteman et al., 2014 for the PCAP definition, but more explanation of Whiteman et al.'s 4.04 MJ/m2 number would be useful.
6. Pg 3. Ln 23. What were the dD and d18O values of the standards, and did they bracket the range of observed delta values?
7. Pg 3 ln 28 / Section 2.2. Please comment on the reproducibility of the calibrations and robustness of the calibration correction. Is it a linear or non-linear correction, both (over certain [H2O] ranges)? There is also no statement regarding instrument precision in the deltas. There is no statement about uncertainty analysis for d-excess (as a function of water vapor concentration). Figure 10 is the only part of the paper that indicates an uncertainty analysis was conducted.
8. Pg. 5. Ln 28. What amount of fossil fuel (for CH4 for example) would be required to produce 500 ppm CDV? It would be helpful to provide this information to put the numbers into context. Figure 1 shows isohumes from 100-500 ppmv, but I don't know if this range of CDV is what contributes to the SLV boundary layer on average or if it's an upper limit estimate. You could frame this is the context of CO2 emissions. Hestia CO2 is available for SLC, so you could estimate what average CDV mole fractions would be on a non-PCP day (using ef = 1-2), and then make estimates of PCAP CDV contributions assuming 24+ hours of emissions accumulate within a lower (average observed PCAP) boundary layer.
9. Pg. 6. Ln. 4. What are the expected ef values, and why?
10. Pg. 6. Ln 9. What type of linear fitting routine is used here? There is error in the x and y variables presumably, which should be accounted for in the fitting.
11. Pg. 7. Ln 10. Again, cross correlations were determined with what kind of fitting routine? There is error in both x and y, although in this case, the error would be much higher in d-excess than CO2. It also would be useful to report in a table the correlations observed during PCAP periods and non-PCAP periods in addition to those reported for the four winters.
12. Pg. 9 ln. 4. During PCAP events, is there an average observed decrease in d-excess per ppm increase in CO2? What magnitude of CO2 enhancement is required to observe a change in d-excess (at the d-excess LOD)?
13. Pg. 9 Ln 12. What about the deposition of vapor to a snow or ice-covered surface when RH w.r.t. ice is 100%. In the presence of ice/snow, would the deposition of vapor result in drier air with a more negative d-excess value? This effect would be more important at night as temperatures fall?
14. Pg. 9 Ln 28-30. This is true, but the measurements you present were all from winter months. There is EIA fossil fuel consumption data available which provides information about the distribution of fossil fuel types consumed for regions in the US at monthly(?) resolution. You surely can make some educated guess about the fossil fuel consumption-weighted emission factor for SLC during winter months.
15. Pg. 11 Fig 5. It is difficult to distinguish between the circles and squares in Figure 5. Could you try larger markers, or filled vs unfilled markers, or circles vs crosses?

16. Pg. 14 Fig 7 (and Figures 8, 9). Can you change the color scale to one that goes from red-purple. It would be easier to distinguish the PCAP periods in the (a) d-excess vs q plots where the PCAP observations track with the CDV moistening lines.
17. Pg. 20 Figure 10 caption. This is the first time that measurement uncertainty is discussed. You report that the shading reflects the standard error, but there is no quantitative discussion of d-excess uncertainty. This should appear in the Methods.
18. Pg. 21 ln 7. This is an indicator that this phenomenon is difficult to observe (even in SLC). This would be an appropriate place to discuss whether the CDV d-excess measurement precision is good enough to observe CDV d-excess in other cities (that may be naturally more humid).
19. Pg 21. Ln 19-20. This is a repeat of one of my comments above, but what about deposition of vapor in ice supersaturated conditions? Is there snow on the ground during this study period? I think this would impart a more negative d-excess value in the remaining vapor.
20. Pg. 21. Ln 24. This is another repeat of one of my earlier comments. I think reporting CDV d-excess ranges are fine for ef=1 or ef=2, but I think you could also make an educated guess based on Patarasuk et al., 2016, EIA, and other literature to say if you believe ef is closer to 1 or 2 (probably closer to 1?). This shows that the community needs information about the partitioning of the fossil fuels consumed in various cities, at the very least at seasonal resolution.
21. Pg. 22. Ln 2. What type of refinements?
22. Pg 22 ln 8. The statement regarding the lack of a robust relationship b/n CDV or CO2 and mixing height refers to the entire wintertime period, or just PCAP events?
23. Pg. 22 Conclusions. The single conclusions paragraph is essentially a summary. Please provide a discussion about the impacts of your work from a broader perspective. Can these studies only be done in wintertime in semi-arid environments? What refinements would advance this science? Where are improvements needed?

Technical Corrections:
1. Pg 2. Ln 3. VSMOW abbreviation not defined
2. Pg 2. Ln 6. "produce" not "product"?
3. Pg 2 Ln 33. VHD abbreviation should appear on previous line after first instance of "valley heat deficit"
4. Pg 4. Ln 3. Meteorological*
5. Pg. 13 ln 25. Remove "a" between "likely" and "due"

---

## Referee Comment (RC2) · I. Levin (Referee) · 5 Mar 2018

Review of the manuscript by Fiorella et al. "Detection and variability of combustion-derived vapor in an urban basin"

**General Remarks:**

The manuscript presents follow-up work and an extension of continuous observations of co-located measurements of atmospheric $CO_2$ mole fraction and deuterium excess in atmospheric water vapor to investigate the impact of combustion-derived water vapor (CDV) at a monitoring station in the Salt Lake City basin in Utah, USA, during winter. The particularly low deuterium excess values of CDV significantly influence the isotopic signature of atmospheric water vapor during inversion situations at cold temperatures when atmospheric humidity is generally low and during stable weather conditions when combustion-derived emissions ($CO_2$ and $H_2O$) accumulate in the atmospheric boundary layer. The authors estimate, in a Keeling-style mixing model approach, the range of CDV d-excess values, which turns out to be large. They claim that these results could be used to constrain contributions of combustion to urban humidity and meteorology (Abstract), or possibly verify $CO_2$ emissions amounts and/or emissions reductions (Conclusions).

From their four-year observations, the authors convincingly show that the isotopic signature of atmospheric water vapor can be significantly modified by CDV during winter, but I am not convinced that there is a realistic chance to use the observed relation between high $CO_2$ and low d-excess in atmospheric water vapor in a quantitative way. As discussed by the authors, the variability of combustion material and its large range of $H_2O$/$CO_2$ stoichiometry when burned to $CO_2$ and $H_2O$ as well as potential isotope effects during production and emission strongly modify d-excess of CDV. Furthermore, not all $CO_2$ emissions during winter can be solely associated with combustion processes, but some $CO_2$ emissions may also originate from biogenic sources that are not associated with net $H_2O$ emissions. Therefore, the constraints on urban humidity and $CO_2$ emissions mentioned in the Abstract and Conclusions, to my understanding are not justified. A sensitivity study including a thorough uncertainty analysis would be required to support these optimistic statements.

In view of the weaknesses of the "tracer" CDV d-excess, I think the manuscript is too detailed. It has too many figures showing similar, mainly semi-quantitative, features that make the manuscript unnecessarily lengthy. For example, I am not sure that all three case studies (described in Figures 7, 8, 9) need to be presented and discussed in detail. Figure 7 would be sufficient to convince the reader that the processes introduced before really take place and are visible in the observations. In addition, Figures 4, 5 and 6 give somewhat redundant information, with Figure 5, to me, being the most convincing. Figure 4 more or less summarizes what is visible in detail in the time series shown in Figure 3, and Figure 6 somehow "hides" the large variability in the Keeling plots, that are expected because the signature of CDV in not well defined and variable in time. I am missing the error analysis that quantifies the ranges of d-excess and emission factors stated.

Technical: (1) There are many abbreviations used in the manuscript (VHD, PCAP, WBB, SDM, ...), which are new for the reader. It would very much help to spell them out again if they had not been used for a while. (2) Please note that $CO_2$ concentrations are also calibrated as micromole per mole (or ppm), but not ppmv (Fig. 7, 8, 9).

**Specific Remarks:**

Introduction, first sentence: please give reference.

Page 2 line 5: "produce";
line 16: "from"

Page 3 Eq. (1): why sum up to 2200 m?

line 22: how long was the tubing and was it heated (e.g. to avoid condensation effects)?

line 25: give reference to script.

Page 4 line 2: how often was calibrated? measurements uncertainties?

line 3: "meteorological";

line 27: is the time shift between ASB and WBB taken into account in the pre-2014 data?

line 28: better spell out CDV in the title.

Page 6 Fig. 1: the yellow line is not well visible;

line 1: is the total $\Delta CO_2$ from combustion processes, i.e. no flux from biosphere?

line 2: subscripts "obs"

Page 7 Fig. 2 and line 6: In the figure (mixing heights) ground level starts at 0 m while in the text total heights in m a.s.l. are reported; this is confusing

Page 9 line 6: is the correlation really "strong" and does this Figure provide new information compared to Fig. 3?

lines 10-14: in Fig. 5, $q_d$ is plotted vs. $q$, the text explanations are thus unclear.

Page 12 Figure 6: would like to see single events here to better judge on the significance of the correlation (see general comment concerning the significance of the Keeling approach to estimate end members)

Page 13: please give times as local station time or in UTC; panels in Figure 7 (and 8, 9) seem to have been mixed up and do not correspond to the text. As d-excess is shown only in relation with moisture (and not vs. time), it is difficult to see the temporal correlations between $CO_2$ and d. Perhaps add a seventh panel.

Pages 15-18: please explain why it is important to discuss these two case studies.

Page 19 line 7: what is MST?

Page 20 Fig. 10: uncertainties hardly visible

Page 21 lines 6-8: May be WBB is generally not well located on the topographic bench; what is SBI?

Page 22 lines 2 and 26-27: this seems to me far from realistic – please justify and make an uncertainty estimate (see general comments)

---

## Author Comment (AC1) · 10 May 2018

**REVIEWER #1**

**This study evaluates a Keeling-style approach for determining the deuterium-excess signature of combustion derived water vapor (CDV) in the Salt Lake City area. The new approach is consistent with values reported in the group's earlier paper, Gorski et al., 2015. The paper also develops criteria for filtering observational periods when atmospheric conditions are most conducive to the accumulation of CDV. These criteria could be used as a starting point for similar studies conducted in other cities. This is the first study that reports multiple years of water vapor isotope measurements to study CDV. While the study is certainly novel, and the quantification of CDV is important, I think the authors could improve the paper by (1) explicitly stating why this study is important using detailed examples, (2) discussing the broader impacts of the work (how does this work further the field, and where else are improvements needed), and (3) providing quantitative support to put the results of the study into context. For example, the reasons some parameter values are used (e.g. emission factors (ef) from 1-2, CDV mole fractions ranging from 100-500 ppm) should be supported with more explanation. The paper is well-written and concise. The specific suggestions listed below, if incorporated, will provide readers with greater context for interpreting the results of the study.**

**Specific Comments:**
1. **Pg. 1. Ln 21. This might be the only sentence in the paper that explicitly states why quantifying CDV emissions is important. Could you expand this idea by detailing possible CDV impacts in urban areas, e.g. impacts on downwind clouds/weather, link between enhanced humidity/temperatures and heat stroke/fatalities in at-risk groups (elderly, sick), influence on photochemistry/aerosol, etc.**

*This is a great suggestion – we've added a few sentences at this point to expand on the complex relationships between CDV, atmospheric stability and meteorology, and potential impacts for human health.*

2. **Pg. 2. Ln 11. This is an appropriate place to introduce the idea of the SLV's seasonally shifting fossil fuel use (and $H_2O:CO_2$ combustion stoichiometry), which adds to the complexity of quantifying CDV emissions. Furthermore, fossil fuel use trends differ from city to city. Describing the complexities of (1) CDV isotope measurements and (2) uncertainties regarding stoichiometry, fossil fuel consumption, and the impacts on CDV d-excess and emissions estimates bolsters your statements regarding the need for refinements to the method (last sentence of abstract). It would also help to communicate the novelty of these types of studies, and the need to continue work in this area.**

*To address this comment, we have added a new section (section 2) between the introduction and methods section that describes fossil fuel use in the SLV and outlines stoichiometric relationships between CDV and $CO_2$ in emissions. We hope that this section both provides additional context necessary for this study, but also provides some useful guidelines for how to estimate emissions factors for other cities.*

*We have appended the text of the added section here to the end of this comment.*

3. **Pg. 2. Ln 16-19. The last line of the introduction indicates that an objective of this study is to investigate relationships involving CDV amount. Does your analysis allow you to report CDV contributions to the SLC boundary layer (Gorski et al reports up to 13% CDV), or do you mean to say your approach allows for the estimation of CDV mole fractions (based on CDV moistening lines in Figure 1, 7-9), or do you mean to say this study intends to report general relationships between atmospheric stability and CDV amount (not necessarily quantitative estimates). Please clarify.**

*The most direct approach for us to follow is the first one suggested by the reviewer. In our initial submission, we did not make quantitative estimates as there were large uncertainties in $d_{CDV}$. The revised approach for estimating* ef *suggested in this review allows for a more meaningful estimation of the CDV contribution. We will be including a table listing maximum $CO_2$ concentration, minimum d-excess compositions, Keeling regression parameters, and estimated CDV fractions using a few different assumptions about the d-excess value in the absence of local emissions with our revised manuscript.*

4. **Pg 3 ln 4. Why is 2200 msl used in the VHD equation? Is it because that's roughly the height of the mountains surrounding the SLV? Or does it have to do with average mixing height (1290 m + 1500 m = 2790 m, so maybe not?)**

*The column integral ends at 2200 m ASL because this is roughly the elevation of the Oquirrh Mountain ridgeline bounding the west end of the SLV. Whiteman et al. (2014) also suggested this elevation as it maximizes the correlation between the VHD metric and PM2.5 concentrations. We have added a sentence indicating that the upper bound of this summation arises from the height of the Oquirrh Mountains:*

The upper bound in the VHD calculation (2200 m) is determined by the elevation of the Oquirrh Mountain ridgeline, which forms the western valley boundary.

5. **Pg 3. Ln 6. You reference Whiteman et al., 2014 for the PCAP definition, but more explanation of Whiteman et al.'s 4.04 MJ/m$^2$ number would be useful.**

*We have added a few sentences here to provide context for this 4.04 MJ/m$^2$ threshold:*

This VHD threshold of 4.04 MJ/m$^2$ defined by Whiteman et al. (2014) corresponds to the mean VHD in days where the SLV daily fine particulate matter concentration (PM$_{2.5}$) exceeds half of the US National Ambient Air Quality Standard for PM2.5 (17.5µg m$^{-3}$). This threshold has been used in subsequent studies of SLV air quality and atmospheric stability (Baasandorj et al., 2017; Bares et al., 2018), and we have retained this convention for intercomparison with prior studies.

6. **Pg 3. Ln 23. What were the dD and d18O values of the standards, and did they bracket the range of observed delta values?**

*Four standards were used throughout this period of record, with a swap in standards made on February 16, 2017 (e.g., the new standards only apply to the last ~10 measurement days of this study. A summary table is provided below:*

| | Prior to February 16, 2017 | | After February 16, 2017 | |
|---|---|---|---|---|
| | $\delta^{18}O$ | $\delta^2H$ | $\delta^{18}O$ | $\delta^2H$ |
| Light standard | -16.0 | -121.0 | -15.88 | -199.66 |
| Heavy standard | -1.23 | -5.51 | 1.65 | 16.9 |

*They did not bracket the range of observed delta values, but we have reason to believe that the potential uncertainty introduced by this situation is small as the Picarro instruments are extremely linear. Internal measurements of VSLAP, which has isotopic compositions of -55.50‰ for $\delta^{18}O$ and -427.5‰ for $\delta^2H$, are within a few tenths of a permil of the values predicted using a calibration based on these standards despite a notably lighter isotopic composition that is significantly lighter than any vapor observed in our study.*

7. **Pg 3 ln 28 / Section 2.2. Please comment on the reproducibility of the calibrations and robustness of the calibration correction. Is it a linear or non-linear correction, both (over certain [H2O] ranges)? There is also no statement regarding instrument precision in the deltas. There is no statement about uncertainty analysis for d-excess (as a function of water vapor concentration). Figure 10 is the only part of the paper that indicates an uncertainty analysis was conducted.**

*We have revised this section of the manuscript to more explicitly describe uncertainties in the data and in our data processing routines, and to provide an estimate of analytical precision. This section has been revised to read as follows:*

Calibration of raw instrument values at ~1 Hz on the instrument scale to hourly averages on the VSMOW scale proceeds across three stages: (1) Measured isotope values are corrected for an apparent dependence on cavity humidity, using correction equations developed by operating the standards delivery module at a range of injection rates, corresponding to cavity humidity values from 500-30000 ppm. Instrumental precision is determined in this step, with uncertainties arising both from a decrease in instrument precision with decreasing cavity humidity, and uncertainty in the regression equation to correct for this bias. The humidity correction is determined by a linear regression of the deviation of isotopic composition from the measured isotopic composition at a reference humidity against the inverse of cavity humidity. The reference humidity used is 17,000-23,000 ppm, which is the typical humidity that liquid water samples are measured and at which the lab standards are calibrated. Additional details on this correction are provided in a supplement. (2) Analyzer measurements are calibrated to the VSMOW-VSLAP scale using two standards of known isotopic composition delivered by the standards delivery module, using calibration periods that bracket a series of ambient vapor measurements to correct for analytical drift, (3) corrected measurements were aggregated to an hourly time step. Measurement uncertainties are primarily limited by changes in instrument precision with cavity humidity, and

$1\sigma$ uncertainties range from 0.88‰ for $\delta^{18}O$, 3.61‰ for $\delta^2H$, and 7.93‰ for d-excess at 1,000 ppm; to 0.14‰ for $\delta^{18}O$, 0.53‰ for $\delta^2H$, and 1.24‰ for d-excess at 10,000 ppm.

*We will include additional details, as well as plots of our correction showing the decrease in precision with decreases in humidity, in a supplement.*

8. **Pg. 5. Ln 28. What amount of fossil fuel (for CH₄ for example) would be required to produce 500 ppm CDV? It would be helpful to provide this information to put the numbers into context. Figure 1 shows isohumes from 100-500 ppmv, but I don't know if this range of CDV is what contributes to the SLV boundary layer on average or if it's an upper limit estimate. You could frame this in the context of CO₂ emissions. Hestia CO₂ is available for SLC, so you could estimate what average CDV mole fractions would be on a non-PCAP day (using ef = 1-2), and then make estimates of PCAP CDV contributions assuming 24+ hours of emissions accumulate within a lower (average observed PCAP) boundary layer.**

*Following the estimated ef value for SLV estimated using the HESTIA dataset and described in our response to point #2 above, we have added a sentence here that translates these CDV concentrations into equivalent CO₂ increases:*

Assuming a representative ef value of 1.5 (section 2), 100 or 500 ppm of CDV correspond to an increase in CO₂ of 66.7 or 333.3 ppm, respectively.

9. **Pg. 6. Ln. 4. What are the expected ef values, and why?**

*We've added a section on likely ef by fuel source in our revisions and found that a reasonable SLV-scale ef value for SLV winter of 1.5. Emissions factors can range from ~0.5-2 though, depending on fuel source, as described in section 2 of our revised manuscript (see point #2). We've clarified our approach to ef in this regression by adding the following sentence:*

The *ef* parameter depends on the molar ratios of hydrogen to carbon in the fuel source; we estimate a fuel-source-weighted SLV-scale *ef* value for winter of 1.5, but note that *ef* values for hydrocarbon fuels can vary from < 0.5 – 2.

10. **Pg. 6. Ln 9. What type of linear fitting routine is used here? There is error in the x and y variables presumably, which should be accounted for in the fitting.**

*It is true that there is measurement error in both x and y. The x component is calculated as a difference of two CO₂ measurements, which each have an estimated uncertainty associated with them of 0.1 ppm. Assuming the errors in these two measurements are uncorrelated, the net uncertainty in the measurements on the x-axis are approximately 0.14 ppm, or 1.4x10⁻⁴ mmol/mol. We estimate that error in the y-axis is primarily determined from the isotope measurement uncertainties, which depend on humidity and range from 1.2‰ at 10000 ppm H₂O to 7.9‰ at 1000 ppm H₂O. Based on this formulation, the y error is >8500x larger than the x error; therefore, we suggest that ordinary least squares fitting is sufficient here.*

**11. Pg. 7. Ln 10. Again, cross correlations were determined with what kind of fitting routine? There is error in both x and y, although in this case, the error would be much higher in d- excess than $CO_2$. It also would be useful to report in a table the correlations observed during PCAP periods and non-PCAP periods in addition to those reported for the four winters.**

*We're a little confused by this comment, as measures of correlation are not sensitive to measurement error. We calculated cross-correlation values using the Pearson definition of the correlation coefficient as the covariance of x and y at lag $\tau$ divided by the product of the standard deviations of x and y:*

$$\rho_{xy}(\tau) = \frac{\text{cov}(x, y)(\tau)}{\sigma_x \sigma_y}$$

*Neither of these values are sensitive to normally-distributed measurement error, and therefore, the cross-correlation value should not be sensitive to differences in measurement error between x and y. This result is in contrast to regression slopes and intercepts, which are sensitive to differences in measurement error between x and y, as typical least-squares regression assumes that all of the measurement error is contained within y.*

**12. Pg. 9 ln. 4. During PCAP events, is there an average observed decrease in d-excess per ppm increase in $CO_2$? What magnitude of $CO_2$ enhancement is required to observe a change in d-excess (at the d-excess LOD)?**

*This is essentially the slope of the linear model presented in Figure 6, following an appropriate scaling of the x-axis from mmol/mol to ppm or µmol/mol. The slope of the best-fit linear mixed model is -268 ± 26 (‰ mmol $H_2O$)/mmol $CO_2$, which corresponds to a slope of -179±17‰ / mmol $CO_2$ assuming the emissions factor of 1.5 that we determined from the HESTIA emissions inventory. This suggests a ~0.18±0.02‰ decrease in d-excess for every ppm increase in $CO_2$. Assuming a $1\sigma$ uncertainty of d-excess of 2.4‰ at 4 mmol/mol humidity (a representative mean DJF value for the SLV), and a considering a $2\sigma$ change to be the LOD, we estimate a ~27 ppm enhancement of $CO_2$ is required to see a measurable change in d-excess.*

*We have added the following sentences detailing this analysis to the end of this section, after we present the regression results in figure 6:*

Based on this regression, we estimate that d-excess decreases by 0.18±0.02 for every ppm increase in $CO_2$. Instrumental precision ($1\sigma$) for d-excess is estimated to be 2.4‰ at the mean DJF humidity value of 4 mmol/mol, implying that enrichments of ~27 ppm $CO_2$ can be detected at the $2\sigma$ level. This estimated detection limit will likely decrease as instrument precision and calibration routines are improved.

**13. Pg. 9 Ln 12. What about the deposition of vapor to a snow or ice-covered surface when RH w.r.t. ice is 100%. In the presence of ice/snow, would the deposition of vapor result in drier air with a more negative d-excess value? This effect would be more important at night as temperatures fall?**

*This is an interesting possibility that we did not discuss in the initial submission. Deposition of snow or ice would have opposing impacts on vapor d-excess depending on whether deposition is occurring at saturation or at supersaturation. Vapor deposition at RH = 100% should raise d-excess, not lower it (e.g., Figure 1 and Galewsky et al., 2011; Jouzel & Merlivat, 1984), but vapor deposition at supersaturation would introduce a kinetic effect that would lower vapor d-excess relative to its equilibrium value at saturation (e.g., Galewsky et al., 2011; Jouzel & Merlivat, 1984). We don't have any direct observations of supersaturation but cannot rule out the possibility of supersaturation on snow surfaces or during cloud formation.*

*If kinetic isotope fractionation during vapor deposition were responsible for the observed decreases in d-excess, we might expect to see temporal coherence between decreases in specific humidity and decreases in d-excess. Instead, we see little change in d-excess overnight while q is decreasing, but strong decreases in d-excess associated with increases in $CO_2$ in the early morning (revised Fig. 10, attached to our response to point 17 below).*

*We've haven't made any changes to the manuscript at this point in the methods, but have included this possibility as a discussion point (see our response to comment #19 below).*

**14. Pg. 9 Ln 28-30. This is true, but the measurements you present were all from winter months. There is EIA fossil fuel consumption data available which provides information about the distribution of fossil fuel types consumed for regions in the US at monthly(?) resolution. You surely can make some educated guess about the fossil fuel consumption- weighted emission factor for SLC during winter months.**

*In our revised version, we've used the emission estimates from the HESTIA dataset (Gurney et al., 2012; Patarasuk et al., 2016) and a simplifying assumption about the fuels corresponding to each sector used in the HESTIA dataset in order to make a more informed estimate of the $H_2O:CO_2$ emissions factor. We find that an* ef *value of 1.5 is appropriate based on the distribution of fuel use across Salt Lake County. We also reassessed our model selection and found more support for a model allowing a random effect in both the slope and intercept. The best fit slope in the new model with an ef of 1.5 is -179±17‰.*

*In light of these changes, we've revised these sentences at pg. 9, L. 24-30 to read:*

The best-fit slope of a linear mixed model allowing for random variation in the slope and intercepts across PCAP events yields an estimate of $d_{CDV}$ of -179±17‰ (assuming ef = 1.5). This estimate of $d_{CDV}$ is consistent with the upper limit of the theoretical estimates and pilot measurements from Gorski et al. (2015), and can be validated by a comprehensive survey of fuels in the SLV.

*We have also changed the appropriate section of the methods to reflect this change in model selection (pg 6, L 6-11):*

We apply two linear mixed models where PCAP-to-PCAP event-scale variability is treated as a

random effect to estimate $d_{CDV}$: in the first, the slope is assumed to be constant across all PCAP events but the intercept is allowed to vary, while in the second, both the slope and intercept are allowed to vary across PCAP events. These models are constructed to find the best-fit slope, and therefore the best-fit estimate of $d_{CDV}$, across all PCAP events. As a result, they implicitly assume that changes in $d_{CDV}$ through time are small compared to changes in $d_{bg}q_{bg}$, or that changes in the emissions profile and components of SLV are small compared to environmental variability in humidity and d-excess. We consider only the second model in our results as we find it has more support than the first model, with selection determined based on lower AIC and BIC scores for the second model.

**15. Pg. 11 Fig 5. It is difficult to distinguish between the circles and squares in Figure 5. Could you try larger markers, or filled vs unfilled markers, or circles vs crosses?**

*We have revised this figure to change the opacity of the circles and squares to help distinguish between PCAP and non-PCAP periods. PCAP periods are high-opacity triangles, while non-PCAP periods are low-opacity circles. These changes have made this figure significantly more readable. The revised figure has been pasted below.*

[Figure]

**16. Pg. 14 Fig 7 (and Figures 8, 9). Can you change the color scale to one that goes from red- purple. It would be easier to distinguish the PCAP periods in the (a) d-excess vs q plots where the PCAP observations track with the CDV moistening lines.**

*The color scale of figures 7 and 8 have been changed to one that spans orange-red-purple, as suggested. Figure 9 and its associated section has been removed following a suggestion from reviewer #2, and to keep the manuscript concise in light of the added sections on combustion*

*stoichiometry and uncertainty analysis. We have also changed the panels in these plots to be (a) $CO_2$ concentration, (b) d-excess, (c) specific humidity (q), and (d) qd vs q to help clarify relationships between $CO_2$, d-excess, and q. The revised figure 7 is included below to illustrate these changes.*

[Figure]

**17. Pg. 20 Figure 10 caption. This is the first time that measurement uncertainty is discussed. You report that the shading reflects the standard error, but there is no quantitative discussion of d-excess uncertainty. This should appear in the Methods.**

*We have revised our manuscript to include a more systematic error analysis, and clarify sources of error (e.g., measurement based, or arising from uncertainty in the regressions). The error shown in our original submission was the standard error of the regression, not of the data underlying the regression. To make the uncertainty in the diurnal cycles more apparent, we have revised figure 10 to include information on data uncertainty. Mean values for each hour across all four years are shown as a black dot, with $1\sigma$ variability shown as a vertical black line. As in the initially submitted version, lines show a GAM estimation of the diurnal cycle to show differences in the diurnal cycle across years, with shading indicating the standard error of the model fit. We have updated the figure caption to reflect these changes.*

*A quantitative discussion of d-excess uncertainty has been added to the methods and is detailed in our response to comment #7 above.*

**18. Pg. 21 ln 7. This is an indicator that this phenomenon is difficult to observe (even in SLC). This would be an appropriate place to discuss whether the CDV d-excess measurement precision is good enough to observe CDV d-excess in other cities (that may be naturally more humid).**

*We view this result as likely reflecting the changing footprint integrated by these measurements, and that the heating emissions were more likely kept lower in the valley. We've added a sentence here that clarifies that this relationship might be more strongly observed elsewhere in the valley, as diurnal cycles of $CO_2$ are also more pronounced elsewhere in the valley (e.g., Mitchell et al., 2018):*

Average diurnal cycles in d-excess and $CO_2$ showed little change overnight outside of PCAP events (Fig. 10), which was unexpected as heating emissions continued throughout the evening. The absence of overnight d-excess and $CO_2$ changes was likely a result of the UOU's location on a topographic bench away from large residential areas, or due to injection of cleaner air from above if a surface-based inversion occurs at an elevation below the UOU site. Long-term records of $CO_2$ have also been collected in lower-elevation areas of the SLV and exhibit a greater buildup of $CO_2$ overnight during the winter (e.g., Mitchell et al., 2018), which suggests that a stronger trend in nighttime d-excess and $CO_2$ values might be observed elsewhere in the SLV.

*We have also added a paragraph to the end of the discussion providing some guidance as to where else this technique might be useful:*

This technique for measuring water from combustion in urban areas can be adapted beyond the SLV, though different environments will present distinct challenges. The SLV is well-suited to detecting the buildup of CDV as it has a dry climate, features a large urban area in a topographic basin, and experiences frequent multi-day periods of high atmospheric stability in the winter. The CDV signal is largest in dry regions or during winter (Fig. 1), and CDV may comprise a larger fraction of urban humidity in these cities for a given level of emissions intensity. However, though the CDV signal is higher at low humidities, instrumental precision is lower. Therefore, at current instrumental precision limits, there is a trade-off between precision of the CDV estimates and the size of the CDV signal.

**19. Pg 21. Ln 19-20. This is a repeat of one of my comments above, but what about deposition of vapor in ice supersaturated conditions? Is there snow on the ground during this study period? I think this would impart a more negative d-excess value in the remaining vapor.**

*Based on our revised figure 10, which includes diurnal cycles in specific humidity, we view the potential role of vapor deposition under supersaturation to be small, but we cannot rule it out. We consider this possibility to have a likely small impact as we see a much closer association in diurnal cycles of d-excess with diurnal cycles of $CO_2$ than of q (Fig. 10, pasted below). Nonetheless, this is a really interesting possibility and we cannot rule it out – we have added the following sentences to this section:*

Deposition of vapor onto ice in supersaturated conditions can also promote a decrease in vapor d-excess (Galewsky et al., 2011; Jouzel & Merlivat, 1984). While we do not have any direct observations of supersaturated conditions, we cannot rule out the possibility of supersaturated conditions occurring when snow is in the valley or during cloud formation. However, we expect that any potential role for vapor deposition under supersaturated conditions on vapor d-excess to be small, as we do not typically observe decreases in d-excess concurrent with decreases in specific humidity (Fig. 10).

**20. Pg. 21. Ln 24. This is another repeat of one of my earlier comments. I think reporting CDV d-excess ranges are fine for ef=1 or ef=2, but I think you could also make an educated guess based on Patarasuk et al., 2016, EIA, and other literature to say if you believe ef is closer to 1 or 2 (probably closer to 1?). This shows that the community needs information about the partitioning of the fossil fuels consumed in various cities, at the very least at seasonal resolution.**

*We have taken this suggestion and made a quantitative estimate of* ef *for winter Salt Lake County using the HESTIA data set* (Gurney et al., 2012; Patarasuk et al., 2016)*. We provide a most likely estimate of 1.5 as a valley-scale* ef *value. A more detailed answer to this point is provided above in our responses to points 8 and 9.*

*In light of this, we have rephrased this sentence as follows:*

We have made an estimate of 1.5 for *ef* through a detailed accounting of emissions or fuel sources from the HESTIA dataset (Patarasuk et al., 2016), but several sources of uncertainty in net *ef* remain. For example, heat exchangers designed to improve heating efficiency may reduce the $H_2O$ concentration in emissions, and potentially alter $d_{CDV}$ as well through condensation of water in the emissions stream (Fig. 1).

**21. Pg. 22. Ln 2. What type of refinements?**

*We have revised this sentence to be more specific:*

Further refinements in CDV detection with stable water vapor isotopes, such as from direct measurements of fuel sources across a range of combustion systems and improved water vapor isotope analyzer precision, may increase the applicability of this method of analyzing fossil fuel emissions, and help verify that background $CO_2$ measurements are free from local emissions.

**22. Pg 22 ln 8. The statement regarding the lack of a robust relationship b/n CDV or $CO_2$ and mixing height refers to the entire wintertime period, or just PCAP events?**

*We've removed this sentence as it could be read ambiguously and the rest of the paragraph conveys our point here. A true quantitative relationship between mixing height and CDV/$CO_2$ amounts is difficult to evaluate with the data we have for a few reasons: (a) atmospheric soundings at the airport occur before sunrise and around sunset every day, and therefore, are*

*unable to capture diurnal changes in mixing height well, and (b) the build-up of $CO_2$ and CDV in the boundary layer requires prolonged stability, not just stability. In this view, evaluating the relationship between current mixing height and $CO_2$ may be misleading.*

**23. Pg. 22 Conclusions. The single conclusions paragraph is essentially a summary. Please provide a discussion about the impacts of your work from a broader perspective. Can these studies only be done in wintertime in semi-arid environments? What refinements would advance this science? Where are improvements needed?**

*We have revised our conclusion to address these points, in accordance with the suggested revisions.*

**Technical Corrections:**

**1. Pg 2. Ln 3. VSMOW abbreviation not defined**

*We have defined "VSMOW" prior to its first use on page 2 in the revised version.*

**2. Pg 2. Ln 6. "produce" not "product"?**

*We were referring to reaction products here, but recognize this sentence was needlessly ambiguous and confusing. We have revised this sentence to read:*

The reaction of $^{18}O$-enriched oxygen with $^2H$-depleted fuels produces vapor with an unusually negative deuterium excess value (d = $\delta^2H$ - $8\delta^{18}O$; Dansgaard, 1964) that is distinct in the "natural" hydrological cycle.

**3. Pg 2 Ln 33. VHD abbreviation should appear on previous line after first instance of "valley heat deficit"**

*We have made this change.*

**4. Pg 4. Ln 3. Meteorological\***

*This typo has been corrected.*

**5. Pg. 13 ln 25. Remove "a" between "likely" and "due"**

*This typo has been corrected.*

**REVIEWER #2 – Dr. Ingeborg Levin**

**Review of the manuscript by Fiorella et al. "Detection and variability of combustion-derived vapor in an urban basin"**

**General Remarks:**

The manuscript presents follow-up work and an extension of continuous observations of co-located measurements of atmospheric $CO_2$ mole fraction and deuterium excess in atmospheric water vapor to investigate the impact of combustion-derived water vapor (CDV) at a monitoring station in the Salt Lake City basin in Utah, USA, during winter. The particularly low deuterium excess values of CDV significantly influence the isotopic signature of atmospheric water vapor during inversion situations at cold temperatures when atmospheric humidity is generally low and during stable weather conditions when combustion-derived emissions ($CO_2$ and $H_2O$) accumulate in the atmospheric boundary layer. The authors estimate, in a Keeling-style mixing model approach, the range of CDV d-excess values, which turns out to be large. They claim that these results could be used to constrain contributions of combustion to urban humidity and meteorology (Abstract), or possibly verify $CO_2$ emissions amounts and/or emissions reductions (Conclusions).

From their four-year observations, the authors convincingly show that the isotopic signature of atmospheric water vapor can be significantly modified by CDV during winter, but I am not convinced that there is a realistic chance to use the observed relation between high $CO_2$ and low d-excess in atmospheric water vapor in a quantitative way. As discussed by the authors, the variability of combustion material and its large range of $H_2O/CO_2$ stoichiometry when burned to $CO_2$ and $H_2O$ as well as potential isotope effects during production and emission strongly modify d-excess of CDV. Furthermore, not all $CO_2$ emissions during winter can be solely associated with combustion processes, but some $CO_2$ emissions may also originate from biogenic sources that are not associated with net $H_2O$ emissions. Therefore, the constraints on urban humidity and $CO_2$ emissions mentioned in the Abstract and Conclusions, to my understanding are not justified. A sensitivity study including a thorough uncertainty analysis would be required to support these optimistic statements.

In view of the weaknesses of the "tracer" CDV d-excess, I think the manuscript is too detailed. It has too many figures showing similar, mainly semi-quantitative, features that make the manuscript unnecessarily lengthy. For example, I am not sure that all three case studies (described in Figures 7, 8, 9) need to be presented and discussed in detail. Figure 7 would be sufficient to convince the reader that the processes introduced before really take place and are visible in the observations. In addition, Figures 4, 5 and 6 give somewhat redundant information, with Figure 5, to me, being the most convincing. Figure 4 more or less summarizes what is visible in detail in the time series shown in Figure 3, and Figure 6 somehow "hides" the large variability in the Keeling plots, that are expected because the signature of CDV in not well defined and variable in time. I am missing the error analysis that quantifies the ranges of d-excess and emission factors stated.

**Technical: (1) There are many abbreviations used in the manuscript (VHD, PCAP, WBB, SDM, …), which are new for the reader. It would very much help to spell them out again if they had not been used for a while.**

*We've revised the text to remove abbreviations that are infrequently used (e.g., SBI, SDM, etc.) – abbreviations that are frequently used remain.*

**(2) Please note that $CO_2$ concentrations are also calibrated as micromole per mole (or ppm), but not ppmv (Fig. 7, 8, 9).**

*We've changed all instances of ppmv to ppm to avoid confusion.*

**Specific Remarks:**
**Introduction, first sentence: please give reference.**

*We have clarified here that this is estimated from carbon emissions as follows:*

Fossil fuel combustion releases carbon dioxide and water to the atmosphere. Annual carbon emissions are estimated to be 9.5 Pg C/y (Le Quéré et al., 2018), which suggests annual water emissions from combustion of ~21.1 Pg, assuming a mean molar ratio of $H_2O:CO_2$ in emissions of 1.5 (section 2, and also Gorski et al., 2015).

**Page 2 line 5: "produce";**

*We were referring to reaction products here, but recognize this sentence was needlessly ambiguous and confusing. We have revised this sentence to read:*

The reaction of $^{18}O$-enriched oxygen with $^2H$-depleted fuels produces vapor with an unusually negative deuterium excess value (d = $\delta^2H$ - $8\delta^{18}O$; Dansgaard, 1964) that is distinct in the "natural" hydrological cycle.

**line 16: "from"**

*Good catch – thanks. This typo has been corrected.*

**Page 3 Eq. (1): why sum up to 2200 m?**

*The column integral ends at 2200 m asl because this is roughly the elevation of the Oquirrh Mountain ridgeline bounding the west end of the SLV. Whiteman et al. (2014) also suggested this elevation as it maximizes the correlation between the VHD metric and PM2.5 concentrations. We have added a sentence indicating that the upper bound of this summation arises from the height of the Oquirrh Mountains:*

The upper bound in the VHD calculation (2200 m) is determined by the elevation of the Oquirrh Mountain ridgeline, which forms the western valley boundary.

**line 22: how long was the tubing and was it heated (e.g. to avoid condensation effects)?**

*The sampling tubing was ~10 m long (half indoors) and was not heated. We observed no condensation in the tubing, and observed no periods of unusual bias between humidity values measured by the Picarro CRDS and the meteorological station.*

**line 25: give reference to script.**

*A link to the processing scripts is provided in the code and data availability section of this manuscript.*

**Page 4 line 2: how often was calibrated? measurements uncertainties?**

*Calibrations were performed every 12 hours, with two standard waters being measured for at least 15 minutes each. Measurement uncertainties are primarily limited by changes in instrument precision with cavity humidity, and $1\sigma$ uncertainties range from 0.88‰ for $\delta^{18}O$, 3.61‰ for $\delta^2H$, and 7.93‰ for d-excess at 1000 ppm; to 0.14‰ for $\delta^{18}O$, 0.53‰ for $\delta^2H$, and 1.24‰ for d-excess at 10000 ppm.*

*We have added this information to the methods section, and also will include plots detailing how uncertainties change with humidity as a supplement.*

**line 3: "meteorological";**

*This typo has been corrected.*

**line 27: is the time shift between ASB and WBB taken into account in the pre-2014 data?**

*We did not shift the ASB data, as the time magnitude of the shift was small and the measurement period where these observations overlapped did not cover an entire annual cycle. We've added the following sentence to clarify this point:*

We do not adjust the ASB time series as the potential time shift is small, and the period of overlapping records is short and does not span a full annual cycle.

**line 28: better spell out CDV in the title.**

*We have made this change.*

**Page 6 Fig. 1: the yellow line is not well visible;**

*We have revised figure 1 to use a gradient of reds to make the 100 ppm CDV isohume more visible. The revised figure is copied here as well.*

[Figure]

**line 1: is the total ΔCO2 from combustion processes, i.e. no flux from biosphere?**

*For Salt Lake City, the biogenic contribution to $\Delta CO_2$ has been shown to be negligible compared to the anthropogenic flux* (Pataki et al., 2003, 2006, 2007; Strong et al., 2011). *We have added the following sentence to this paragraph to clarify this point:*

Observations of urban $\delta^{13}C\text{-}CO_2$ and atmospheric modeling of the SLV indicate that wintertime increases in $CO_2$ above background concentrations are driven by anthropogenic emissions, and that the contribution from local respiration to urban $CO_2$ enhancement is likely negligible (Pataki et al., 2003, 2005, 2007; Strong et al., 2011).

**line 2: subscripts "obs"**

*Good catch – this typo has been corrected.*

**Page 7 Fig. 2 and line 6: In the figure (mixing heights) ground level starts at 0 m while in the text total heights in m a.s.l. are reported; this is confusing**

*We have revised the sentence at line 6 to express heights in meters above ground level, making this sentence consistent with Fig. 2:*

Calculated mixing heights ranged from the surface (0 m AGL) to 3390 m AGL, with a median value of 270 m AGL.

**Page 9 line 6: is the correlation really "strong" and does this Figure provide new information compared to Fig. 3?**

*We agree that Figure 4 in our original submission did not provide any data that was not already presented in Figures 3 or 5.*

*Therefore, we have removed this figure in our revisions.*

**lines 10-14: in Fig. 5, qd is plotted vs. q, the text explanations are thus unclear.**

*Good catch – we've revised this section to read as follows to indicate we're analyzing a plot of qd vs q:*

Changes in the product of *q* and d-excess relative to *q* from atmospheric moistening and drying processes in the absence of CDV are expected to follow a linear relationship with a positive slope (Fig. 1). In contrast, addition of CDV to the atmosphere will promote strong, linear, and negative-sloped deviations from this *qd-q* relationship that are proportional to the amount of CDV. These patterns are observed in our measurements, where *qd* values trend up with *q* at low $CO_2$ concentrations, and decrease linearly with increasing $CO_2$ (Fig. 5).

**Page 12 Figure 6: would like to see single events here to better judge on the significance of the correlation (see general comment concerning the significance of the Keeling approach to estimate end members)**

*We have taken several steps to hopefully improve our implementation of the Keeling approach, described below:*

*1)* *We have gone back and assessed whether the model in the original submission represents the best model formulation. We have determined that it was not. Our original submission featured a linear-mixed model, where a random effect across PCAP events was allowed in the intercept. In our revisions, we have discovered that a model fit allowing for random effects in both the slope and intercept across PCAP events has more support via lower AIC and BIC scores. To illustrate this change, we have provided a revised figure 6 and revised text in the methods:*

[Figure]

2) *We have provided summary statistics (e.g., slope with regression uncertainty and an $R^2$ value) for the Keeling approach as a supplementary table for each individual PCAP event using a more simple, linear ordinary least squares model.*

3) *Following suggestions from reviewer #1, we have made a more quantitative estimate of the ef parameter to narrow the ranges of $d_{CDV}$ estimated through this regression. To generate an improved estimate of ef, we used the HESTIA data set (Patarasuk et al., 2016), which is a bottom-up emissions inventory at hourly and building-scale resolution and breaks down emissions by economic sector. We estimate that at the valley scale, an emissions weighted ef value of 1.5 is appropriate.*

**Page 13: please give times as local station time or in UTC; panels in Figure 7 (and 8, 9) seem to have been mixed up and do not correspond to the text. As d-excess is shown only in relation with moisture (and not vs. time), it is difficult to see the temporal correlations between CO2 and d. Perhaps add a seventh panel.**

*The times provided are in UTC; we have changed "Z" to UTC to clarify this.*

**Pages 15-18: please explain why it is important to discuss these two case studies.**

*We sought to investigate compare a few different PCAP scenarios, and how the d-excess and $CO_2$ timeseries coevolved under different conditions. However, in light of the expanded discussion on SLV fuel sources and stoichiometry and the additional uncertainty analysis, we have removed the third case study presented in our initial submission for brevity. We have*

*decided to retain the first two as they show different patterns, with the former showing a strong coupling between d-excess and CO₂, and the latter illustrating a period where though there is strong diurnal variability between d-excess and CO₂, changes in specific humidity seem to be largely driven by other factors.*

**Page 19 line 7: what is MST?**

*MST is "Mountain Standard Time," the local time. We have added a definition for this abbreviation before it is first used in this instance.*

**Page 20 Fig. 10: uncertainties hardly visible**

*We've made three changes to Figure 10 to help interpret the uncertainty in these panels. First, we've made the uncertainties in the GAM fits more prominent to better show the error in the model fits. Second, we've added a layer to this plot indicating the variability in the data these models are constructed on. Mean hourly $\Delta$d-excess and $\Delta CO_2$ values are shown as black dots with $1\sigma$ variability shown as vertical lines. Third, as differences across months are small, we have plotted these quantities as seasonal averages instead of monthly averages. Monthly plots, like those in Fig. 10 of the original submission, will be included as supplementary material.*

*In response to comments raised by reviewer #1, we've also added a column in this figure showing the diurnal cycle of specific humidity. The above steps to clarify uncertainty are also extended to this column.*

**Page 21 lines 6-8: Maybe WBB is generally not well located on the topographic bench; what is SBI?**

*We've clarified the role of the WBB on the topographic bench here, and how it may contribute to the patterns we observe over night:*

Average diurnal cycles in d-excess and CO₂ showed little change overnight outside of PCAP events (Fig. 10), which was unexpected as heating emissions continued throughout the evening. The absence of overnight d-excess and CO₂ changes was likely a result of the UOU's location on a topographic bench away from large residential areas, or due to injection of cleaner air from above if a surface-based inversion occurs at an elevation below the UOU site. Long-term records of CO₂ have also been collected in lower-elevation areas of the SLV and exhibit a greater buildup of CO₂ overnight during the winter (e.g., Mitchell et al., 2018), which suggests that a stronger trend in nighttime d-excess and CO₂ values might be observed elsewhere in the SLV.

**Page 22 lines 2 and 26-27: this seems to me far from realistic – please justify and make an uncertainty estimate (see general comments)**

*After consideration, we agree that this conclusion is too optimistic. Instead, we suggest that water isotope observations may be useful to help validate whether CO₂ observations that are taken to represent "background" values, as background CO₂ values should not be significantly*

*correlated with water vapor d-excess if these $CO_2$ values are not influenced by local emissions. We have revised the sentence at line 2 to read:*

Further refinements in CDV detection with stable water vapor isotopes, such as from direct measurements of fuel sources across a range of combustion systems and improved water vapor isotope analyzer precision, may increase the applicability of this method of analyzing fossil fuel emissions, and help 
[revised manuscript text omitted]

---

## Author Comment (AC2) · 10 May 2018

This additional comment contains two new tables (as a supplement) that were requested by the reviewers: Table 2, which shows Keeling-style slope estimates for each PCAP event, and Table 3, which makes explicit estimates of the fraction of urban humidity arising from combustion for each PCAP event with paired water isotope and $CO_2$ data.

We also referenced a revised version of Fig. 10 in our response to the reviewers, but this file was not included in this comment. We are including it here.

Please also note the supplement to this comment:

[Figure]

https://www.atmos-chem-phys-discuss.net/acp-2017-1106/acp-2017-1106-AC2-supplement.pdf

[Figure]

[Figure]

Figure 10. Seasonal average diurnal cycles of Δd-excess (left column), $\Delta CO_2$ (center column), and Δq (right column) for days in PCAP conditions (top row) or non-PCAP conditions (bottom row). The diurnal cycle is approximated here as the deviation from a 24-hr moving average. Mean values across all four years are shown as black symbols, with black vertical lines indicating 1σ variability. The mean diurnal cycle is modeled for each year independently as a GAM using cubic cyclic smoothing splines, and regression standard error shown as a color shading. The influence of CDV in the diurnal cycle is apparent from comparing Δd-excess and $CO_2$ cycles: increases in $CO_2$ co-occur with decreases in d-excess during the early morning and late afternoon periods.

**Fig. 1.**

**Supplement:**

Table 2. Estimated Keeling-style slopes and coefficients of determination for each PCAP event

| Start | End | Keeling Slope (ef = 1.5) | Keeling $R^2$ |
|---|---|---|---|
| 10-Dec-13 12 | 14-Dec-13 00 | -190 | 0.33 |
| 15-Dec-13 12 | 19-Dec-13 12 | -259 | 0.77 |
| 26-Dec-13 00 | 29-Dec-13 00 | -275 | 0.62 |
| 30-Dec-13 12 | 31-Dec-13 12 | -89 | 0.17 |
| 02-Jan-14 12 | 1-Jan-14 00 | -101 | 0.13 |
| 17-Jan-14 00 | 22-Jan-14 12 | -173 | 0.3 |
| 24-Jan-14 12 | 26-Jan-14 12 | -185 | 0.34 |
| 31-Dec-14 12 | 3-Jan-15 12 | -134 | 0.42 |
| 7-Jan-15 12 | 11-Jan-15 00 | -241 | 0.34 |
| 15-Jan-15 12 | 17-Jan-15 00 | -228 | 0.59 |
| 12-Jan-16 12 | 14-Jan-16 00 | -128 | 0.25 |
| 22-Jan-16 12 | 23-Jan-16 12 | -199 | 0.55 |
| 28-Jan-16 00 | 29-Jan-16 00 | -206 | 0.15 |
| 8-Feb-16 12 | 14-Feb-16 12 | -25 | 0.001 |
| 20-Dec-16 00 | 21-Dec-16 00 | -130 | 0.06 |
| 27-Dec-16 12 | 28-Dec-16 12 | -45 | 0.005 |
| 29-Dec-16 12 | 2-Jan-17 00 | -193 | 0.52 |
| 7-Jan-17 12 | 8-Jan-17 12 | -189 | 0.34 |
| 14-Jan-17 12 | 15-Jan-17 12 | -379 | 0.64 |
| 18-Jan-17 00 | 19-Jan-17 00 | -41 | 0.44 |
| 29-Jan-17 12 | 2-Feb-17 12 | -232 | 0.08 |
| 13-Feb-17 12 | 15-Feb-17 12 | -328 | 0.62 |

Table 3. Estimates of urban humidity fraction arising from fossil fuel combustion in the SLV across different PCAP periods

| Start | End | Max $CO_2$ | Min $d_{obs}$ | $d_{nat}$ before PCAP | $d_{nat}$, last value with < 425 ppm $CO_2$ | $q_{CDV}/q_{obs}$, $d_{nat}$ before PCAP | $q_{CDV}/q_{obs}$, last value with < 425 ppm $CO_2$ |
|---|---|---|---|---|---|---|---|
| 10-Dec-13 12 | 14-Dec-13 00 | 558.0 | -7.0±2.3 | 20.8±2.8 | 20.8±2.8 | 13.9±2.9 | 13.9±2.9 |
| 15-Dec-13 12 | 19-Dec-13 12 | 632.9 | -10.9±2.0 | 10.0±2.0 | 11.4±1.8 | 11.1±2.4 | 11.7±2.3 |
| 26-Dec-13 00 | 29-Dec-13 00 | 592.4 | -13.8±1.9 | 6.3±2.2 | 6.3±2.2 | 10.8±2.5 | 10.8±2.5 |
| 30-Dec-13 12 | 31-Dec-13 12 | 557.6 | -4.1±1.8 | 5.7±2.1 | 6.9±2.1 | 5.3±2.2 | 5.9±2.2 |
| 02-Jan-14 12 | 1-Jan-14 00 | 555.7 | -8.1±1.6 | -1.3±1.7 | 1.7±1.7 | 3.8±1.9 | 5.4±1.9 |
| 17-Jan-14 00 | 22-Jan-14 12 | 569.0 | -9.6±1.8 | 1.8±1.7 | 3.4±1.9 | 6.3±2.0 | 7.1±2.2 |
| 24-Jan-14 12 | 26-Jan-14 12 | 511.6 | -7.8±2.2 | -1.6±2.3 | 2.1±1.8 | 3.5±2.6 | 5.5±2.3 |
| 31-Dec-14 12 | 3-Jan-15 12 | 500.3 | -10.5±2.6 | 5.6±5.4 | 5.6±5.4 | 8.7±4.5 | 8.7±4.5 |
| 7-Jan-15 12 | 11-Jan-15 00 | 580.6 | -3.6±1.3 | 5.2±1.4 | 9.6±2.1 | 4.8±1.5 | 7.0±1.9 |
| 15-Jan-15 12 | 17-Jan-15 00 | 525.5 | 2.2±2.0 | 9.7±2.3 | 11.6±2.1 | 4.0±2.3 | 4.9±2.2 |
| 12-Jan-16 12 | 14-Jan-16 00 | 526.5 | -5.9±2.2 | -2.3±3.0 | 2.5±2.9 | 2.0±3.0 | 4.6±2.9 |
| 22-Jan-16 12 | 23-Jan-16 12 | 524.9 | -4.3±1.9 | 2.5±2.6 | 3.2±2.4 | 3.7±2.5 | 4.1±2.4 |
| 28-Jan-16 00 | 29-Jan-16 00 | 490.5 | -3.4±2.1 | 2.1±2.1 | -3.2±2.5 | 3.0±2.3 | 0.1±2.6 |
| 8-Feb-16 12 | 14-Feb-16 12 | 532.8 | -2.7±1.9 | 0.7±2.1 | 0.6±3.3 | 1.9±2.2 | 1.8±2.9 |
| 20-Dec-16 00 | 21-Dec-16 00 | 491.1 | -9.8±2.6 | -1.4±3.6 | -10.7±4.3 | 4.7±3.5 | -0.5±4.1 |
| 27-Dec-16 12 | 28-Dec-16 12 | 522.9 | -17.0±2.9 | -10.9±3.2 | -9.8±3.1 | 3.6±3.7 | 4.3±3.6 |
| 29-Dec-16 12 | 2-Jan-17 00 | 652.9 | -23.1±2.3 | -12.1±2.4 | -6.3±2.6 | 6.6±2.9 | 9.7±3.0 |
| 7-Jan-17 12 | 8-Jan-17 12 | 588.2 | -25.9±3.9 | -19.0±3.6 | -17.6±4.5 | 4.3±4.7 | 5.1±5.2 |
| 14-Jan-17 12 | 15-Jan-17 12 | 459.1 | -2.4±1.9 | -0.7±1.8 | 1.4±1.7 | 1.0±2.1 | 2.1±2.0 |
| 18-Jan-17 00 | 19-Jan-17 00 | 512.7 | -4.9±2.3 | -3.4±2.2 | -1.0±2.1 | 0.9±2.6 | 2.2±2.5 |
| 29-Jan-17 12 | 2-Feb-17 12 | 525.8 | -14.7±3.1 | -8.3±3.0 | -5.0±3.2 | 3.7±3.6 | 5.6±3.7 |
| 13-Feb-17 12 | 15-Feb-17 12 | 537.9 | -9.4±2.1 | -1.0±2.3 | 3.1±2.0 | 4.7±2.5 | 6.9±2.4 |

---

## Author Response (AR1)

**REVIEWER #1**

This study evaluates a Keeling-style approach for determining the deuterium-excess signature of combustion derived water vapor (CDV) in the Salt Lake City area. The new approach is consistent with values reported in the group's earlier paper, Gorski et al., 2015. The paper also develops criteria for filtering observational periods when atmospheric conditions are most conducive to the accumulation of CDV. These criteria could be used as a starting point for similar studies conducted in other cities. This is the first study that reports multiple years of water vapor isotope measurements to study CDV. While the study is certainly novel, and the quantification of CDV is important. I think the authors could improve the paper by (1) explicitly stating why this study is important using detailed examples, (2) discussing the broader impacts of the work (how does this work further the field, and where else are improvements needed), and (3) providing quantitative support to put the results of the study into context. For example, the reasons some parameter values are used (e.g. emission factors (ef) from 1-2, CDV mole fractions ranging from 100-500 ppm) should be supported with more explanation. The paper is well- written and concise. The specific suggestions listed below, if incorporated, will provide readers with greater context for interpreting the results of the study.

**Specific Comments:**

1. Pg. 1. Ln 21. This might be the only sentence in the paper that explicitly states why quantifying CDV emissions is important. Could you expand this idea by detailing possible CDV impacts in urban areas, e.g. impacts on downwind clouds/weather, link between enhanced humidity/temperatures and heat stroke/fatalities in at-risk groups (elderly, sick), influence on photochemistry/aerosol, etc.

This is a great suggestion – we've added a few sentences at this point to expand on the complex relationships between CDV, atmospheric stability and meteorology, and potential impacts for human health. This section now reads (pg 2, L. 1-7):

In turn, water vapor from fossil fuel combustion may impact urban air quality and meteorology, including through direct changes in radiative balance by increased water vapor concentrations (Holmer and Eliasson, 1999; McCarthy et al., 2010), impacts on aerosols and cloud properties (Pruppacher and Klett, 2010; Mölders and Olson, 2004; Kourtidis et al., 2015; Twohy et al., 2009; Carlton and Turpin, 2013; Kaufman and Koren, 2006), and altered local or downwind precipitation amounts (Rosenfeld et al., 2008). Where combined with atmospheric stratification, these changes can potentially lengthen or intensify periods of elevated particulate pollution in cities, which would directly impact public health through increased incidence of acute cardiovascular (Morris et al., 1995; Brook et al., 2010) or respiratory (Dockery and Pope, 1994) illness.

2. Pg. 2. Ln 11. This is an appropriate place to introduce the idea of the SLV's seasonally shifting fossil fuel use (and H2O:CO2 combustion stoichiometry), which adds to the complexity of quantifying CDV emissions. Furthermore, fossil fuel use trends differ from city to city. Describing the complexities of (1) CDV isotope measurements and (2) uncertainties regarding stoichiometry, fossil fuel

consumption, and the impacts on CDV d-excess and emissions estimates bolsters your statements regarding the need for refinements to the method (last sentence of abstract). It would also help to communicate the novelty of these types of studies, and the need to continue work in this area.

To address this comment, we have added a new section (section 2) between the introduction and methods section that describes fossil fuel use in the SLV and outlines stoichiometric relationships between CDV and CO2 in emissions. We hope that this section both provides additional context necessary for this study, but also provides some useful guidelines for how to estimate emissions factors for other cities. This section forms page 3 and L1-4 of page 4 in the revised manuscript.

3. Pg. 2. Ln 16-19. The last line of the introduction indicates that an objective of this study is to investigate relationships involving CDV amount. Does your analysis allow you to report CDV contributions to the SLC boundary layer (Gorski et al reports up to 13% CDV), or do you mean to say your approach allows for the estimation of CDV mole fractions (based on CDV moistening lines in Figure 1, 7-9), or do you mean to say this study intends to report general relationships between atmospheric stability and CDV amount (not necessarily quantitative estimates). Please clarify.

The most direct approach for us to follow is the first one suggested by the reviewer. In our initial submission, we did not make quantitative estimates as there were large uncertainties in  $d_{CDV}$ . The revised approach for estimating ef suggested in this revision allows for a more meaningful estimation of the CDV contribution.

We've clarified the goals of this study in the last paragraph of the introduction (L. 23-31 of P. 2), and added sections and tables to the results examining CDV amount relationships.

4. Pg 3 ln 4. Why is 2200 msl used in the VHD equation? Is it because that's roughly the height of the mountains surrounding the SLV? Or does it have to do with average mixing height (1290 m + 1500 m = 2790 m, so maybe not?)

The column integral ends at 2200 m ASL because this is roughly the elevation of the Oquirrh Mountain ridgeline bounding the west end of the SLV. Whiteman et al. (2014) also suggested this elevation as it maximizes the correlation between the VHD metric and PM2.5 concentrations. We have added a sentence indicating that the upper bound of this summation arises from the height of the Oquirrh Mountains:

The upper bound in the VHD calculation (2200 m) is determined by the elevation of the Oquirrh Mountain ridgeline, which forms the western valley boundary.

**5. Pg 3. Ln 6. You reference Whiteman et al., 2014 for the PCAP definition, but more explanation of Whiteman et al.'s 4.04 MJ/m2 number would be useful.**

We have added a few sentences here to provide context for this 4.04 MJ/m2 threshold (P5, L. 7-

10):

This VHD threshold of 4.04  $MJ/m^2$  corresponds to the mean VHD in days where the SLV daily fine particulate matter concentration (PM2.5) exceeds half of the US National Ambient Air Quality Standard for PM2.5 (17.5µg m-3) (Whiteman et al. 2014). This threshold has been used in subsequent studies of SLV air quality and atmospheric stability (Baasandorj et al., 2017; Bares et al., 2018), and we have retained this convention for intercomparison with prior studies.

**6. Pg 3. Ln 23. What were the dD and d18O values of the standards, and did they bracket the range of observed delta values?**

Four standards were used throughout this period of record, with a swap in standards made on February 16, 2017 (e.g., the new standards only apply to the last ~10 measurement days of this study. A summary table is provided below:

|                | Prior to Febr  | uary 16, 2017 | After February 16, 2017 |              |  |
|----------------|----------------|---------------|-------------------------|--------------|--|
|                | $\delta^{18}O$ | $\delta^2 H$  | $\delta^{18}O$          | $\delta^2 H$ |  |
| Light standard | -16.0          | -121.0        | -15.88                  | -199.66      |  |
| Heavy standard | -1.23          | -5.51         | 1.65                    | 16.9         |  |

They did not bracket the range of observed delta values, but we have reason to believe that the potential uncertainty introduced by this situation is small as the Picarro instruments are extremely linear. Internal measurements of VSLAP, which has isotopic compositions of -55.50‰ for  $\delta^{18}O$  and -427.5‰ for  $\delta^{2}H$ , are within a few tenths of a permil of the values predicted using a calibration based on these standards despite a notably lighter isotopic composition that is significantly lighter than any vapor observed in our study.

7. Pg 3 ln 28 / Section 2.2. Please comment on the reproducibility of the calibrations and robustness of the calibration correction. Is it a linear or non-linear correction, both (over certain [H2O] ranges)? There is also no statement regarding instrument precision in the deltas. There is no statement about uncertainty analysis for d-excess (as a function of water vapor concentration). Figure 10 is the only part of the paper that indicates an uncertainty analysis was conducted.

We have revised this section of the manuscript to more explicitly describe uncertainties in the data and in our data processing routines, and to provide an estimate of analytical precision. This section has been revised to read as follows (P5. L30 to P6, L12):

Calibration of raw instrument values at  $\sim$ 1 Hz on the instrument scale to hourly averages on the VSMOW scale proceeds across three stages: (1) Measured isotope values are corrected for an apparent dependence on cavity humidity, using correction equations developed by operating the standards delivery module at a range of injection rates, corresponding to cavity humidity values from 500-30000 ppm. Instrumental precision is determined in this step, with uncertainties arising both from a decrease in instrument precision with decreasing cavity humidity, and uncertainty in the regression equation to correct for this bias. The humidity correction is determined by a linear

regression of the deviation of isotopic composition from the measured isotopic composition at a reference humidity against the inverse of cavity humidity. The reference humidity used is 15,000-25,000 ppm, which is the typical humidity that liquid water samples are measured and at which the lab standards are calibrated. Additional details on this correction are provided in a supplement. (2) Analyzer measurements are calibrated to the VSMOW-VSLAP scale using two standards of known isotopic composition delivered by the standards delivery module, using calibration periods that bracket a series of ambient vapor measurements to correct for analytical drift, (3) corrected measurements were aggregated to an hourly time step. Measurement uncertainties are primarily limited by changes in instrument precision with cavity humidity, and  $1\sigma$  uncertainties range from 0.88‰ for  $\delta^{18}$ O, 3.61‰ for  $\delta^{2}$ H, and 7.93‰ for d-excess at 1,000 ppm; to 0.14‰ for  $\delta^{18}$ O, 0.53‰ for  $\delta^{2}$ H, and 1.24‰ for d-excess at 10,000 ppm.

**Additional details, as well as plots of our correction showing the decrease in precision with decreases in humidity, are included as a supplement.**

8. Pg. 5. Ln 28. What amount of fossil fuel (for CH4 for example) would be required to produce 500 ppm CDV? It would be helpful to provide this information to put the numbers into context. Figure 1 shows isohumes from 100-500 ppmv, but I don't know if this range of CDV is what contributes to the SLV boundary layer on average or if it's an upper limit estimate. You could frame this in the context of CO2 emissions. Hestia CO2 is available for SLC, so you could estimate what average CDV mole fractions would be on a non-PCAP day (using ef = 1-2), and then make estimates of PCAP CDV contributions assuming 24+ hours of emissions accumulate within a lower (average observed PCAP) boundary layer.

Following the estimated of value for SLV estimated using the HESTIA dataset and described in our response to point #2 above, we have added a sentence here that translates these CDV concentrations into equivalent CO2 increases (P.8, L.7-8):

Assuming a representative of value of 1.5 (section 2), 100 or 500 ppm of CDV correspond to  $CO_2$  increases of 66.7 or 333.3 ppm, respectively.

**9. Pg. 6. Ln. 4. What are the expected ef values, and why?**

We've added a section on likely ef by fuel source in our revisions and found that a reasonable SLV-scale ef value for SLV winter of 1.5. Emissions factors can range from ~0.5-2 though, depending on fuel source, as described in section 2 of our revised manuscript (see point #2). We've clarified our approach to ef in this regression by adding the following sentence (P. 8, L. 17-19):

The *ef* parameter depends on the molar ratios of hydrogen to carbon in the fuel source; we estimate a fuel-source-weighted SLV-scale *ef* value for winter of 1.5, but note that *ef* values for hydrocarbon fuels can vary from < 0.5 - 2.

**10. Pg. 6. Ln 9. What type of linear fitting routine is used here? There is error in the x**

**and y variables presumably, which should be accounted for in the fitting.**

It is true that there is measurement error in both x and y. The x component is calculated as a difference of two CO2 measurements, which each have an estimated uncertainty associated with them of 0.1 ppm. Assuming the errors in these two measurements are uncorrelated, the net uncertainty in the measurements on the x-axis are approximately 0.14 ppm, or  $1.4x10^{-4}$  mmol/mol. We estimate that error in the y-axis is primarily determined from the isotope measurement uncertainties, which depend on humidity and range from 1.2% at 10000 ppm H2O to 7.9‰ at 1000 ppm H2O. Based on this formulation, the y error is >8500x larger than the x error; therefore, we suggest that ordinary least squares fitting is sufficient here.

11. Pg. 7. Ln 10. Again, cross correlations were determined with what kind of fitting routine? There is error in both x and y, although in this case, the error would be much higher in d- excess than CO2. It also would be useful to report in a table the correlations observed during PCAP periods and non-PCAP periods in addition to those reported for the four winters.

We're a little confused by this comment, as measures of correlation are not sensitive to measurement error. We calculated cross-correlation values using the Pearson definition of the correlation coefficient as the covariance of x and y at lag  $\tau$  divided by the product of the standard deviations of x and y:

$$\rho_{xy}(\tau) = \frac{\operatorname{cov}(x, y)(\tau)}{\sigma_x \sigma_y}$$

Neither of these values are sensitive to normally-distributed measurement error, and therefore, the cross-correlation value should not be sensitive to differences in measurement error between x and y. This result is in contrast to regression slopes and intercepts, which are sensitive to differences in measurement error between x and y, as typical least-squares regression assumes that all of the measurement error is contained within y.

**12. Pg. 9 ln. 4. During PCAP events, is there an average observed decrease in d-excess per ppm increase in CO2? What magnitude of CO2 enhancement is required to observe a change in d-excess (at the d-excess LOD)?**

This is essentially the slope of the linear model presented in Figure 6, following an appropriate scaling of the x-axis from mmol/mol to ppm or  $\mu$ mol/mol. The slope of the best-fit linear mixed model is  $-268 \pm 26$  (% mmol H2O)/mmol CO2, which corresponds to a slope of  $-179\pm17\%$  / mmol CO2 assuming the emissions factor of 1.5 that we determined from the HESTIA emissions inventory. This suggests a ~0.18±0.02‰ decrease in d-excess for every ppm increase in CO2. Assuming a 1 $\sigma$  uncertainty of d-excess of 2.4‰ at 4 mmol/mol humidity (a representative mean DJF value for the SLV), and a considering a 2 $\sigma$  change to be the LOD, we estimate a ~27 ppm enhancement of CO2 is required to see a measurable change in d-excess.

We have added the following sentences detailing this analysis to the end of this section, after we present the regression results in figure 6 (P. 12, L. 21-25):

Based on this regression, we estimate that d-excess decreases by  $0.18\pm0.02\%$  for every ppm increase in CO2. Instrumental precision (1 $\sigma$ ) for d-excess is estimated to be 2.4‰ at the mean DJF humidity value of 4 mmol/mol, implying that enrichments of ~40 ppm CDV can be detected at the  $2\sigma$  level. This estimated detection limit will likely decrease as instrument precision and calibration routines are improved.

**13. Pg. 9 Ln 12. What about the deposition of vapor to a snow or ice-covered surface when RH w.r.t. ice is 100%. In the presence of ice/snow, would the deposition of vapor result in drier air with a more negative d-excess value? This effect would be more important at night as temperatures fall?**

This is an interesting possibility that we did not discuss in the initial submission. Deposition of snow or ice would have opposing impacts on vapor d-excess depending on whether deposition is occurring at saturation or at supersaturation. Vapor deposition at RH = 100% should raise d-excess, not lower it (e.g., Figure 1 and Galewsky et al., 2011; Jouzel & Merlivat, 1984), but vapor deposition at supersaturation would introduce a kinetic effect that would lower vapor d-excess relative to its equilibrium value at saturation (e.g., Galewsky et al., 2011; Jouzel & Merlivat, 1984). We don't have any direct observations of supersaturation but cannot rule out the possibility of supersaturation on snow surfaces or during cloud formation.

If kinetic isotope fractionation during vapor deposition were responsible for the observed decreases in d-excess, we might expect to see temporal coherence between decreases in specific humidity and decreases in d-excess. Instead, we see little change in d-excess overnight while q is decreasing, but strong decreases in d-excess associated with increases in  $CO_2$  in the early morning (revised Fig. 10, attached to our response to point 17 below).

We've haven't made any changes to the manuscript at this point in the methods, but have included this possibility as a discussion point (see our response to comment #19 below).

14. Pg. 9 Ln 28-30. This is true, but the measurements you present were all from winter months. There is EIA fossil fuel consumption data available which provides information about the distribution of fossil fuel types consumed for regions in the US at monthly(?) resolution. You surely can make some educated guess about the fossil fuel consumption- weighted emission factor for SLC during winter months.

In our revised version, we've used the emission estimates from the HESTIA dataset (Gurney et al., 2012; Patarasuk et al., 2016) and a simplifying assumption about the fuels corresponding to each sector used in the HESTIA dataset in order to make a more informed estimate of the  $H_2O:CO_2$  emissions factor. We find that an ef value of 1.5 is appropriate based on the distribution of fuel use across Salt Lake County. We also reassessed our model selection and found more support for a model allowing a random effect in both the slope and intercept. The best fit slope in the new model with an ef of 1.5 is -179±17‰.

*In light of these changes, we've revised these sentences at pg. 9, L. 24-30 to read (in revised MS, this section is at p. 12, l. 17-21):*

The best-fit slope of a linear mixed model allowing for random variation in the slope and intercepts across PCAP events yields an estimate of  $d_{CDV}$  of  $-179\pm17\%$  (assuming ef = 1.5). This estimate of  $d_{CDV}$  is consistent with the upper limit of the theoretical estimates and pilot measurements from Gorski et al. (2015), and could be validated by a comprehensive survey of fuels in the SLV.

*We have also changed the appropriate section of the methods to reflect this change in model selection (pg 8, L 21-30):*

We apply two linear mixed models where PCAP-to-PCAP event-scale variability is treated as a random effect to estimate  $d_{CDV}$ : in the first, the slope is assumed to be constant across all PCAP events but the intercept is allowed to vary, while in the second, both the slope and intercept are allowed to vary across PCAP events. These models are constructed to find the best-fit slope, and therefore the best-fit estimate of  $d_{CDV}$ , across all PCAP events. As a result, they implicitly assume that changes in  $d_{CDV}$  through time are small compared to changes in  $d_{bg}q_{bg}$ , or that changes in the emissions profile of SLV are small compared to environmental variability in humidity and d-excess. We consider only the second model in our results as we find it has more support than the first model, with this selection determined based on lower AIC and BIC scores for the second model.

**15. Pg. 11 Fig 5. It is difficult to distinguish between the circles and squares in Figure 5. Could you try larger markers, or filled vs unfilled markers, or circles vs crosses?**

We have revised this figure to change the opacity of the circles and squares to help distinguish between PCAP and non-PCAP periods. PCAP periods are high-opacity triangles, while non-PCAP periods are low-opacity circles. These changes have made this figure significantly more readable. The revised figure has been pasted below.

16. Pg. 14 Fig 7 (and Figures 8, 9). Can you change the color scale to one that goes from red- purple. It would be easier to distinguish the PCAP periods in the (a) d-excess vs q plots where the PCAP observations track with the CDV moistening lines.

The color scale of figures 7 and 8 have been changed to one that spans orange-red-purple, as suggested. Figure 9 and its associated section has been removed following a suggestion from reviewer #2, and to keep the manuscript concise in light of the added sections on combustion stoichiometry and uncertainty analysis. We have also changed the panels in these plots to be (a) temperature, (b) specific humidity, (c) wind speed, (d) CO2 concentration, (e) d-excess, (f) qd vs q to help clarify relationships between CO2, d-excess, and q. The revised figure 7 is included below to illustrate these changes.

**17. Pg. 20 Figure 10 caption. This is the first time that measurement uncertainty is discussed. You report that the shading reflects the standard error, but there is no quantitative discussion of d-excess uncertainty. This should appear in the Methods.**

We have revised our manuscript to include a more systematic error analysis, and clarify sources of error (e.g., measurement based, or arising from uncertainty in the regressions). The error shown in our original submission was the standard error of the regression, not of the data underlying the regression. To make the uncertainty in the diurnal cycles more apparent, we have revised figure 10 to include information on data uncertainty. Mean values for each hour across all four years are shown as a black dot, with  $1\sigma$  variability shown as a vertical black line. As in the initially submitted version, lines show a GAM estimation of the diurnal cycle to show differences in the diurnal cycle across years, with shading indicating the standard error of the model fit. We have updated the figure caption to reflect these changes, and the revised figure is pasted below.